# RNA supply drives physiological granule assembly in neurons

Karl E. Bauer[1], Niklas Bargenda [1], Rico Schieweck[1], Christin Illig[1], Inmaculada Segura[1,2], Max Harner[1] & Michael A. Kiebler [1]✉

Membraneless cytoplasmic condensates of mRNAs and proteins, known as RNA granules, play pivotal roles in the regulation of mRNA fate. Their maintenance fine-tunes time and location of protein expression, affecting many cellular processes, which require complex protein distribution. Here, we report that RNA granules—monitored by DEAD-Box helicase 6 (DDX6)—disassemble during neuronal maturation both in cell culture and in vivo. This process requires neuronal function, as synaptic inhibition results in reversible granule assembly. Importantly, granule assembly is dependent on the RNA-binding protein Staufen2, known for its role in RNA localization. Altering the levels of free cytoplasmic mRNA reveals that RNA availability facilitates DDX6 granule formation. Specifically depleting RNA from DDX6 granules confirms RNA as an important driver of granule formation. Moreover, RNA is required for DDX6 granule assembly upon synaptic inhibition. Together, this data demonstrates how RNA supply favors RNA granule assembly, which not only impacts subcellular RNA localization but also translation-dependent synaptic plasticity, learning, and memory.

[1] BioMedical Center, Dept. Cell Biology and Anatomy, Medical Faculty, Ludwig Maximilians University, Großhaderner Str. 9, 82152 Planegg-Martinsried, Germany. [2] Present address: Max Planck Institute for Biological Intelligence (in foundation), Am Klopferspitz 18, 82152 Martinsried, Germany. ✉email: mkiebler@lmu.de

Eukaryotic cells constantly adapt their gene expression in response to specific cellular demands. This happens primarily at the transcriptional level, however, a number of additional mechanisms allow for subsequent post-transcriptional regulation. Post-transcriptional gene regulation crucially contributes to many important cellular processes, e.g. development, differentiation, and maturation of cells in tissues, but also to many pathological alterations. Prominent examples are cancer and neurodegeneration among many others[1–3]. The nervous system sticks out as it displays by far the highest transcriptomic diversity[4,5]. Hence, to enable spatiotemporally controlled protein expression, RNA-binding proteins (RBPs) play crucial roles by guiding mRNAs through the different stages of their life cycle in the cell[6]. RBPs assemble with their target mRNAs into RNA granules or ribonucleoprotein particles (RNPs)[7]. Since their original discovery[8,9], many different types of RNA granules have been identified and described, e.g. processing bodies (P-bodies), stress granules, and transport granules among others[10–12]. Importantly, the content of these RNPs is dynamically regulated by both the demand and supply of RNAs[13–15]. While the demand crucially depends on translational activity, the supply of RNAs is additionally determined by transcription, splicing, translational repression, and eventually decay. Consequently, RNPs are able to dynamically assemble and disassemble in order to react to changing cellular demands. This behavior has been nicely visualized by time-lapse fluorescence microscopy[16,17]. In addition to their assembly dynamics, RNPs are very heterogenous in protein and RNA composition[7,18]. Therefore, it is tempting to postulate that a network of RBPs and eventually RNA granules serves as a dynamic buffer system to allow homeostatic RNA metabolism inside the cell[6]. This RNA-based mechanism is a prerequisite for cells to respond to different stimuli in order to remodel the neuronal proteome. Thereby, RNPs balance the RNA demand and supply that is needed for adequate protein synthesis, which ultimately enables cellular plasticity. Disrupting the balance of RNA homeostasis can have important consequences, such as the development of neurodegenerative diseases, which in some cases are hallmarked by pathological deposits of RNA granule components[19]. Here, the concept of liquid-liquid phase separation has an important impact on our understanding of how RNA and RBPs form RNA condensates inside cells[20]. Interestingly, RNA recruitment affects the size and composition of RNA-protein condensates in cells[21]. While the relevance of RNA availability for RNA granule assembly[22] has already been studied in vitro and under diseased or stress conditions, its physiological impact is much less understood. Therefore, we aimed to investigate the assembly and disassembly of RNA granules under physiological conditions in cells. We chose the RNA helicase DDX6 (Rck, p54), which has been shown to mediate translational suppression of specific mRNAs in P-bodies[23,24]. Notably, DDX6 is not only found in P-bodies, but also in other types of RNA granules, including stress granules and Staufen2 (Stau2) containing transport RNPs[18]. The yeast homolog Dhh1 regulates RNP assembly and turnover[25]. Together, this underlines the importance of DDX6 in gene expression regulation[18,26]. Here, we report that DDX6 granule formation is regulated by RNA supply in the neuronal cytosol, as well as neuronal maturation and synaptic activity.

## Results

**Cytoplasmic DDX6 granules physiologically disassemble during neuronal maturation**. To get first insight into the regulation of DDX6 granule formation, we took advantage of primary neurons in cell culture as a model system. As hippocampal neurons undergo immense morphological and physiological changes during their maturation[27], we speculated that these alterations might influence RNA granule properties as well, and studied DDX6 granule formation at different stages of maturation[28]. We observed cells with larger and smaller DDX6 granules at all observed time points during maturation, though at different proportions. To quantify this variability in the cell population, we manually counted > 100 cells per time point and biological replicate by eye and categorized cells into two groups: cells containing either preferentially large or small granules. Examples of this quantification are provided in Suppl. Fig. 1a. We found a striking shift in the population from immature hippocampal neurons with large granules to mature neurons with small granules, indicating significant reorganization (Fig. 1a, b; $F_{3,8} = 0.0044$). This occurred in a similar fashion during the maturation of cultured cortical neurons (Suppl. Fig. 1b). To determine whether this effect was specific to DDX6 granules or rather reflected global remodeling during neuronal maturation, we investigated granule formation of 6 other RBPs with diverse functions (i.e. ZBP1, UPF1, RBM14, Pur α, Pum2, Mov10) (Suppl. Fig. 1c). Interestingly, we observed various distinct localization patterns. However, none of these RBPs displayed a change in granule formation comparable to that of DDX6, indicating an RBP specific effect. Detailed single particle quantification of DDX6 revealed a decrease in granule size ($p = 0.0452$; 8 days in vitro (DIV) vs. 29 DIV), but an increase in their number ($p = 0.0081$), further corroborating RNA granule reorganization (Fig. 1c, d). Mature neurons showed a smaller variability in granule size within single cells compared to immature neurons, suggesting that granules were not only smaller but also more homogeneous (Suppl. Fig. 1d). Additionally, we observed a shift of DDX6 immunofluorescence from granules to the cytosol compartment during maturation (Fig. 1e; $p = 0.00096$). The observed disassembly was accompanied by decreased DDX6 protein levels, both in hippocampal (Suppl. Fig. 1e) and cortical neurons (Suppl. Fig. 1f). To confirm that these observations hold true in the living animal, we performed DDX6 immunostainings on brain slices of 8 day old and 10 month old mice. Indeed, we detected a clear reduction in DDX6 granule size in mature compared to young mice, both in the hippocampus and cortex (Fig. 1f, Suppl. Fig. 1g). To gain insight into granule dynamics in living cells, we performed live imaging of green fluorescent protein (GFP) fused to DDX6 (GFP-DDX6) transiently expressed in hippocampal neurons. Granules generally displayed diffusion-like mobility and fused or split in individual instances, providing first insight into how granule assembly can be regulated over a longer time period (Suppl. Fig. 1h and Suppl. Movie 1). This is in line with a recent publication, showing that the assembly of cytoplasmic condensates is regulated by coalescence in vivo[29]. Taken together, our findings clearly show that physiological DDX6 granules are remodeled during neuronal maturation in vivo.

**Neuronal activity regulates DDX6 granule assembly in mature hippocampal neurons**. Next, we asked the question how synaptic activity, a key physiological hallmark of neuronal maturation[28], might alter the properties of DDX6 granules, as previously suggested for other types of RNPs[13,17]. Here, we interfered with synaptic activity by simultaneous inhibition of AMPA receptors, NMDA receptors, and voltage-gated sodium channels through combined application of CNQX, AP5, and TTX[30] (subsequently termed silenced) and investigated its impact on DDX6 granule formation. Interestingly, the inhibition of synaptic activity in mature neurons resulted in the reassembly of DDX6 into larger granules (Fig. 2a, b; $p = 0.0012$). Detailed single particle analysis revealed a significant increase in average granule size and a decrease in the number of granules in the cell body (Fig. 2c;

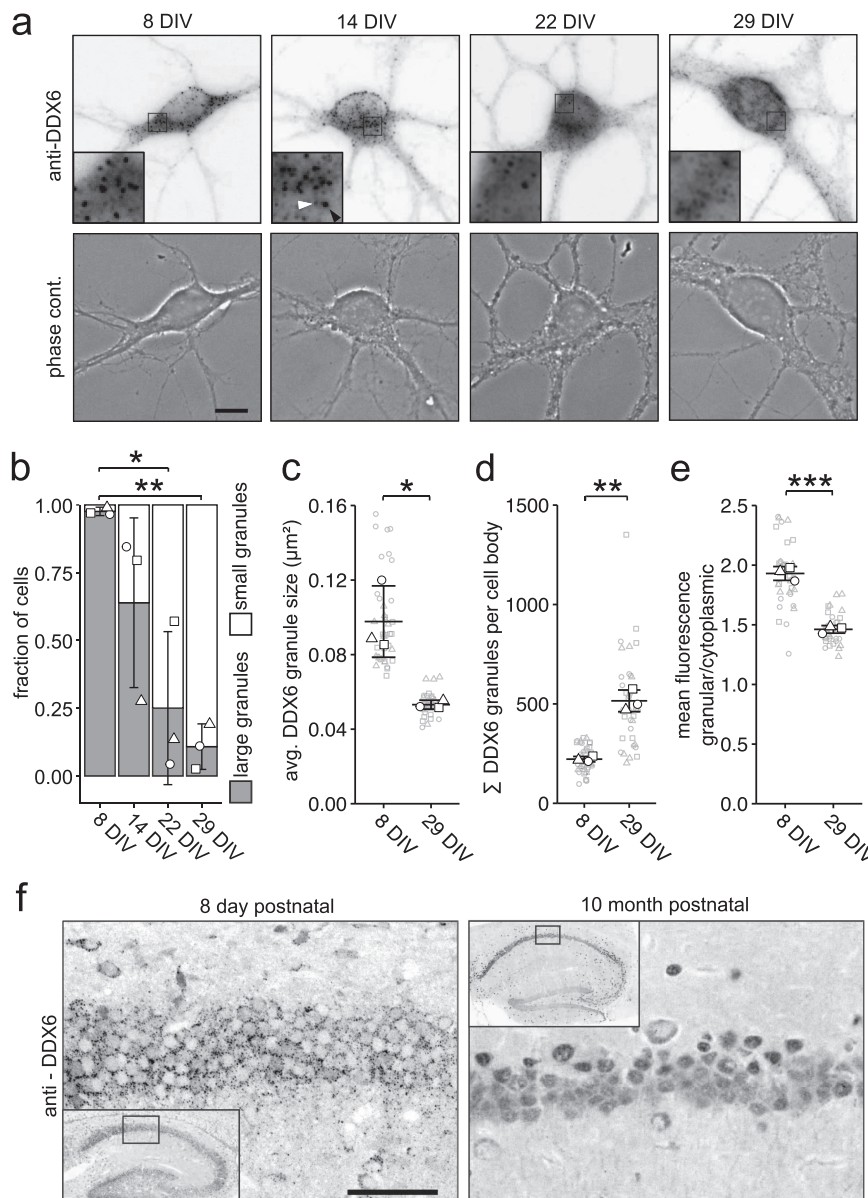

**Fig. 1 Cytoplasmic DDX6 granules disassemble during neuronal maturation in cell culture and in vivo. a** Representative examples of DDX6 immunostaining and phase contrast of 8, 14, 22, and 29 days in vitro (DIV) hippocampal neurons in culture. Boxed regions in images are displayed as magnified insets. White arrowhead indicates representative small granule, black arrowhead indicates representative large granule. Scale bar 10 µm. **b** Bar plot displaying quantification of cell population by fraction of cells containing either large or small DDX6 granules as exemplified in **a**, at 8, 14, 22 and 29 DIV, respectively. Data represents mean ± standard deviation of three independent neuronal cultures. Distinct dot symbols indicate biological replicates. At least 100 cells/condition/experiment were quantified. Asterisks represent p-values obtained by Tukey's test post-hoc to one-way ANOVA analysis (*$p < 0.05$, **$p < 0.01$). $F_{3,8} = 0.0044$. **c, d, e** Dot plots displaying average DDX6 granule size (**c**), total DDX6 granule number (**d**) and granular to cytoplasmic DDX6 fluorescence ratio (**e**) of individual cell bodies in 8 and 29 DIV hippocampal neurons in culture. Small gray symbols represent single cells while larger white symbols indicate the average of each replicate. Horizontal line and error bars represent mean of replicates and standard deviation ($n = 3$ biologically independent experiments). Asterisks represent p-values obtained by two-sided Student's t-test (*$p < 0.05$, **$p < 0.01$, ***$p < 0.001$). $p = 0.0452$ (**c**); $p = 0.0081$ (**d**). **f** DDX6 immunostaining on sagittal brain tissue slices displaying the hippocampus of 8 day and 10 month postnatal mice. Boxed regions in overviews show location of magnified region. Scale bar 50 µm. This experiment was repeated independently 3 times with similar results.

$p = 0.00053$, Fig. 2d; $p = 0.00010$). Importantly, these newly formed assemblies were not stress granules, nor did this treatment induce stress granules (Suppl. Fig. 2a). Reinstating endogenous neuronal activity by wash-off of inhibitors ('recovery') resulted in the quick reversal of granule assembly within fifteen minutes (Fig. 2e; $F_{2,18} = 1.59\text{e-}05$, treatment 1; $F_{2,18} = 2.38\text{e-}11$, treatment 2). Additionally, DDX6 disassembly could be rapidly induced by synaptic activation of NMDA receptors, independent

of the prior treatment. To gain further insight into how granules are disassembled upon NMDA treatment, we transiently expressed GFP-DDX6 in hippocampal neurons and performed live imaging during NMDA application and subsequent NMDA wash-off (Suppl. Fig. 2b and Suppl. Movie 2). We observed DDX6 granules gradually disassemble during NMDA treatment and conversely reassemble upon wash-off, a process distinct to the fusion and splitting of granules observed at baseline neuronal

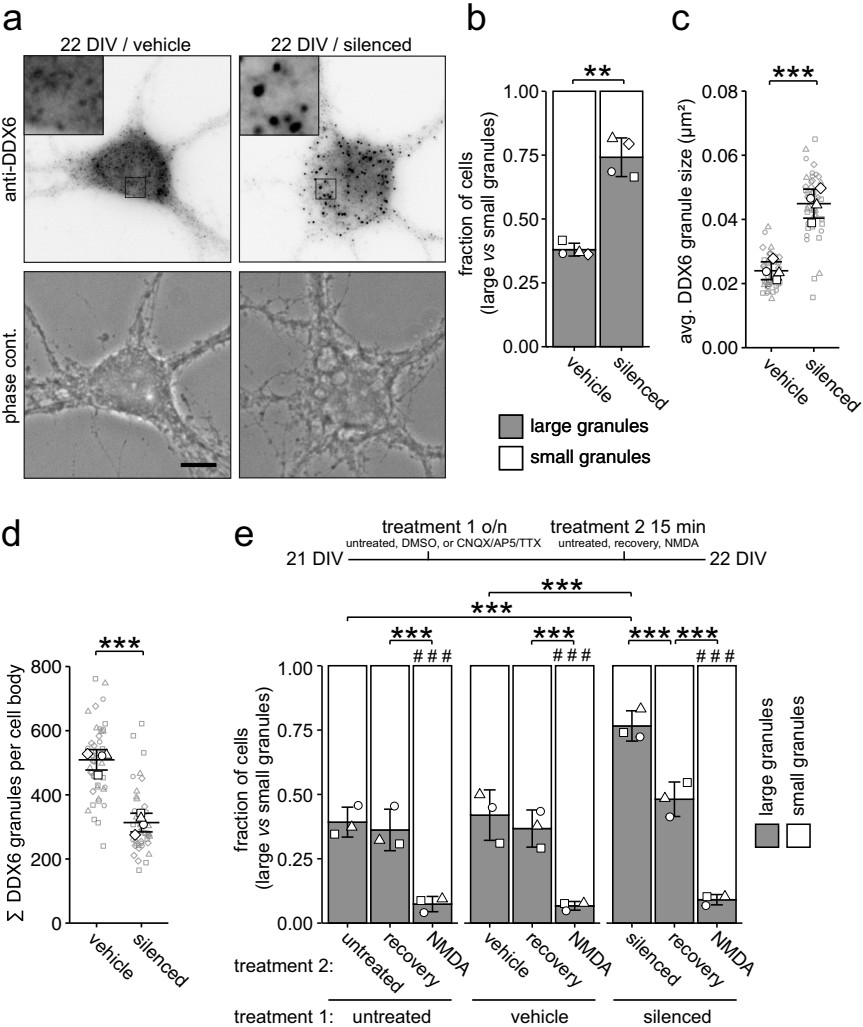

**Fig. 2 Chemical inhibition of neuronal activity selectively regulates the assembly of cytoplasmic DDX6 granules in mature hippocampal neurons.**
**a** Representative examples of DDX6 immunostainings and phase contrast pictures of 22 DIV hippocampal neurons in culture under vehicle (DMSO) treated or silenced (100 μM CNQX, 50 μM AP5, 1 μM TTX) conditions. Boxed regions in images are displayed as magnified insets. Scale bar 10 μm. **b, e** Bar plots displaying quantification of cell population by fraction of cells containing either large or small DDX6 granules as exemplified in **a** under untreated, vehicle treated, or silenced conditions (**b, e**), followed by recovery or NMDA treatment (**e**). Experimental outline is presented in **e**. Data represents mean ± standard deviation of three independent neuronal cultures. Distinct dot symbols indicate biological replicates. At least 100 cells/condition/experiment were quantified. **c, d** Dot plots displaying average DDX6 granule size (**c**) and DDX6 granule number (**d**) of individual cell bodies under vehicle treated (DMSO) or silenced conditions. Small gray symbols represent single cells while larger white symbols indicate the average of each replicate. Horizontal line and error bars represent mean of replicates and standard deviation ($n = 4$ biologically independent experiments). Asterisks represent $p$-values obtained by two-sided Student's $t$-test (**b–d**) or Tukey's test post-hoc to two-way ANOVA analysis (**e**) (**$p < 0.01$). Hashtags represent $p$-values obtained by Tukey's test compared to untreated conditions (**e**) (###$p < 0.001$). $p = 0.0012$ (**b**), $p = 0.00053$ (**c**), $p = 0.00010$ (**d**), $F_{2,18} = 1.59e-05$, treatment 1, $F_{2,18} = 2.38e-11$, treatment 2 (**e**).

activity (Suppl. Fig. 1h and Suppl. Movie 1). Together, these experiments revealed that DDX6 granule assembly is dynamically regulated by synaptic activation or inhibition.

**Stau2 promotes DDX6 granule assembly upon neuronal inhibition.** Synaptic activity crucially determines the localization and translatability of mRNAs[31,32]. Therefore, we speculated that synaptic transmission might modulate DDX6 granule formation by the availability of transcripts. To test whether RBPs involved in RNA transport would supply RNAs needed for DDX6 granule formation, we focused on the double-stranded RBP Stau2, which we have previously identified as an interaction partner of DDX6 in neurons[18]. Stau2 recognizes complex RNA secondary structures predominantly in the 3' untranslated region (UTR) of its targets and is an essential RNA transport protein responsible for

localizing a significant fraction of the neuronal transcriptome within different cellular compartments[30,33–36]. A DEAD box RNA helicase such as DDX6 may play a role in the regulation of such complex targets. Moreover, Stau2 can transiently interact with other types of RNA granules in living neurons[17] suggesting that it serves as a dynamic RNA distributor and regulator in cells. To investigate whether Stau2 indeed associates with DDX6 in the cell body of neurons, we performed co-immunostainings (Fig. 3a). Though Stau2 and DDX6 generally displayed distinct localization patterns, we observed instances where these two proteins clearly colocalized, indicating transient interactions may occur. To test for its functional contribution to the assembly of DDX6 granules in mature neurons, we depleted Stau2 protein from cells using lentiviral transduction of a short-hairpin RNA targeting *Stau2* mRNA[34] (Suppl. Fig. 3a, b). This reduction in

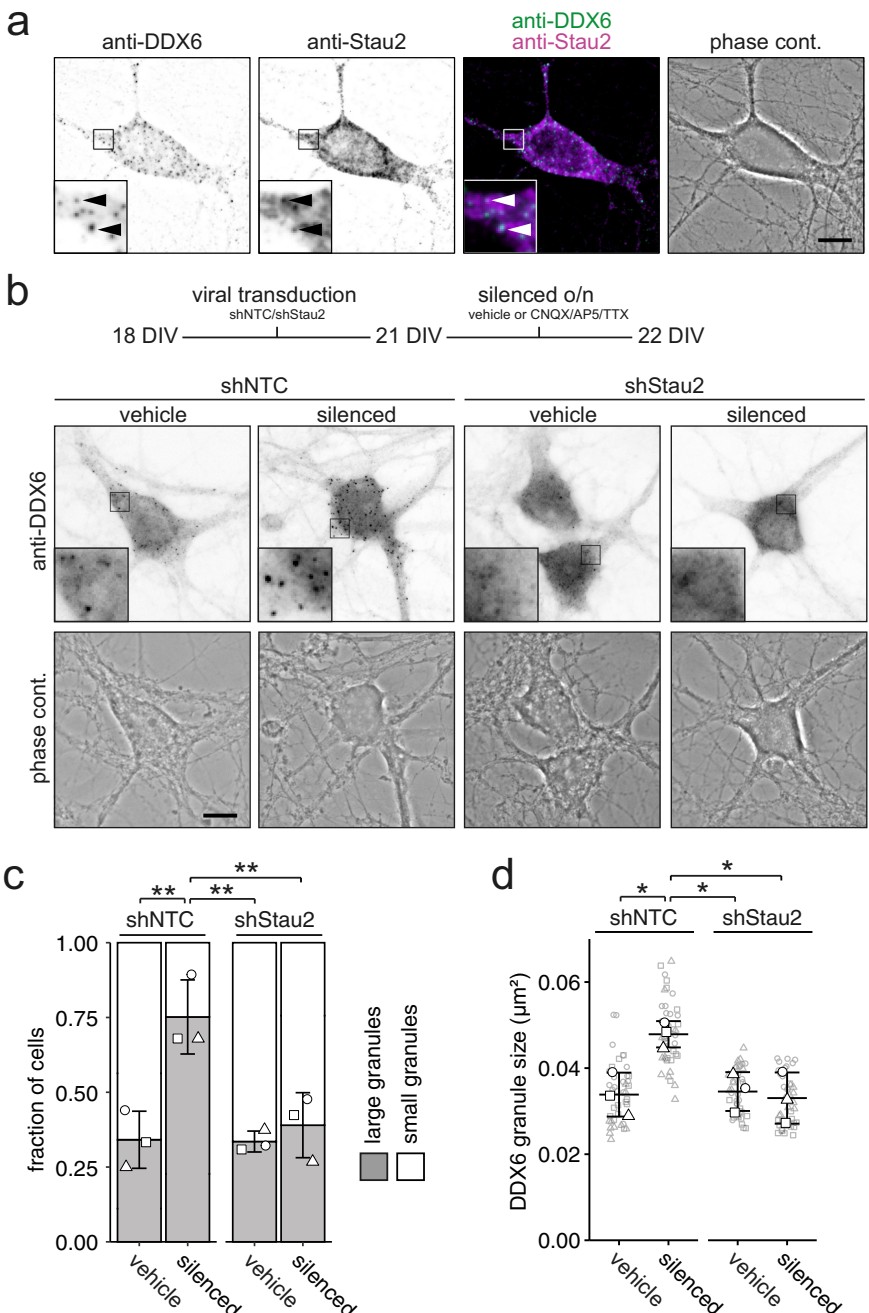

**Fig. 3 Stau2 depletion inhibits DDX6 granule assembly upon neuronal inhibition. a** Representative example of DDX6 and Stau2 immunostaining, merge and phase contrast pictures of a 22 DIV hippocampal neuron. Arrowheads indicate instances of co-localization. **b** Experimental outline and representative examples of DDX6 immunostainings and phase contrast pictures of shNTC and shStau2 transduced 22 DIV hippocampal neurons in culture either under vehicle (DMSO) treated or silenced (100 μM CNQX, 50 μM AP5, 1 μM TTX) conditions. Boxed regions in images are displayed as magnified insets. Scale bars 10 μm. **c** Bar plot displaying quantification of cell population by fraction of cells containing either large or small DDX6 granules as exemplified in **b**. Distinct dot symbols indicate biological replicates. At least 100 cells/condition/experiment were quantified. **d** Dot plot displaying average DDX6 granule size of individual cell bodies. Small gray symbols represent single cells while larger white symbols indicate the average of each replicate. Data represents mean ± standard deviation of three independent neuronal cultures. Asterisks represent *p*-values obtained by Tukey's test post-hoc to two-way ANOVA analysis (*$p < 0.05$, **$p < 0.01$). $F_{1,8} = 0.00316$, Stau2 depletion (**c**); $F_{1,8} = 0.0110$, neuronal inhibition (**c**), $F_{1,8} = 0.0336$, Stau2 depletion (**d**); $F_{1,8} = 0.0530$, neuronal inhibition (**d**).

Stau2 did not alter DDX6 granule size under basal conditions (Fig. 3b, c; $F_{1,8} = 0.00316$, Stau2 depletion; $F_{1,8} = 0.0110$, neuronal inhibition). However, in the absence of Stau2 neuronal inhibition failed to induce DDX6 assembly. This was confirmed by single particle analysis, showing that a significant increase in DDX6 granule size (Fig. 3d, $F_{1,8} = 0.0336$, Stau2 depletion;

$F_{1,8} = 0.0530$, neuronal inhibition) and a reduction in granule number (Suppl. Fig. 3c) upon neuronal inhibition was prevented in Stau2 depleted cells. Together, this data indicates that Stau2 is required for dynamic DDX6 granule assembly upon neuronal inhibition, raising the possibility that the regulation of mRNAs is a determining factor in this process.

**DDX6 granule assembly is facilitated by the supply of cytoplasmic non-translating mRNAs.** Next, we inquired if translational activity, which determines whether an mRNA is covered by ribosomes or accessible for RNA granule formation, affects RNA supply to granules. Translational inhibitors have been shown to affect RNA granule size[37–40]. To test whether RNA availability is indeed crucial for DDX6 assembly in neurons, we exploited three different translation inhibitors: puromycin (PMY), harringtonine (HRN), and cycloheximide (CHX). Though all three compounds inhibit translation, they differ in their underlying mechanism, as PMY dissociates translating ribosomes and releases the bound transcript, HRN blocks initiating ribosomes and thereby induces ribosome runoff, whereas CHX results in stalling and stabilization of ribosomes on mRNAs[41–43]. Thus, all three inhibitors have a different impact on the pool of mRNAs that are available for granule formation; while PMY and, in a translation speed dependent manner, HRN eventually increase the pool, making mRNAs abundant in the cytoplasm and available for other interactors, CHX restricts the number of assembly competent RNAs. We found that PMY treatment resulted in fast DDX6 granule assembly (Fig. 4a, b, $p = 0.02769$). Additionally, HRN treatment caused a gradual assembly upon prolonged incubation (Fig. 4d, e; $F_{3,8} = 0.00333$), possibly due to the induction of ribosome runoff that depends on translational speed (Suppl. Fig. 4a). Detailed single particle analysis of PMY and HRN treated neurons revealed a significant increase in average DDX6 granule size (Fig. 4c; $p = 0.0022$, Fig. 4f; $p = 0.0096$) and decrease in granule number (Suppl. Fig. 4b; $p = 0.0145$, Suppl. Fig. 4c; $p = 0.0273$) in both cases. Conversely, neurons treated with low concentrations of CHX showed a moderate decrease in the neuronal population containing large DDX6 granules (Suppl. Fig. 4d, e). This small effect is likely due to the fact that granules are generally already quite small at baseline in 22 DIV neurons. To better assess the effects of CHX at 22 DIV and to inquire whether it acts upstream or downstream of neuronal activity, we first inhibited neuronal activity to induce large granules and subsequently applied CHX as shown in Fig. 4g. Interestingly, large DDX6 granules induced by neuronal inhibition disassembled when cells were treated with low concentrations of CHX (Fig. 4g, h; $p = 0.00084$), opposing the assembly promoting effect of neuronal inhibition. Again, these assemblies did not represent stress granules, nor did the treatments induce stress granules on their own (Suppl. Fig. 4f). As DDX6 granules assemble upon the supply of cytoplasmic non-translating mRNA, we were curious to know whether DDX6, a well-studied RNA helicase implicated in translation[44], might interact with translating mRNAs or ribosomes. Therefore, we performed polysome profiling in developing and mature cortical neurons (Suppl. Fig. 4g). DDX6 was clearly depleted from polysome fractions in both conditions, indicating that the protein either interacts mainly with non-translating mRNA or only transiently interacts with polysomes. Together, our data demonstrate that balancing RNA availability and translation activity crucially regulate DDX6 granule formation in neurons.

**Depletion of RNA from DDX6 granules reduces their assembly capability.** Having established that RNA availability is essential to promote DDX6 granule assembly in living cells, we investigated whether RNA supply is indeed needed for dynamic DDX6 granule assembly/disassembly. To further investigate the effects of RNA depletion specifically in DDX6 granules in the living cell, we generated a reporter construct consisting of GFP fused to DDX6 (GFP-DDX6), which was C-terminally tagged with the endonuclease RNase1 (Fig. 5a), to effectively deplete RNA from DDX6 granules. We first confirmed that DDX6 granules were accessible

for RNase1 digestion, by exploiting differential centrifugation combined with an in vitro RNase1 treatment (Suppl. Fig. 5a). To verify that the tagged RNase1 was indeed enzymatically active, GFP-DDX6 and GFP-DDX6-RNase1 were expressed in HEK cells, the expressed fusion protein enriched by anti-GFP immunoprecipitation, and incubated with purified RNA. Here, we observed a clear reduction of intact RNA in samples incubated with GFP-DDX6-RNase1 compared to GFP-DDX6 (Suppl. Fig. 5b, c). Next, we transiently transfected GFP-DDX6 or GFP-DDX6-RNase1 reporters into hippocampal neurons (Fig. 5b). Both reporters formed granules that contained DDX6 and most often DCP1a, a protein commonly associated with DDX6[45,46], as shown by immunostaining. Neurons containing the GFP-DDX6-RNase1 reporter showed a small but clear reduction in GFP granule size (Fig. 5c, $p = 0.0306$; Suppl. Fig. 5d) and a non-significant increase in granule number (Fig. 5d), compared to GFP-DDX6 transfected cells. Importantly, these effects resembled the rearrangement of endogenous DDX6 granules observed in neurons during maturation (Fig. 1). Notably, both GFP-DDX6 and GFP-DDX6-RNase1 reporters were expressed at similar levels (Suppl. Fig. 5e). To confirm, that this effect is in fact due to RNA depletion in granules, rather than global degradation, we generated two additional RNase1 reporters that do not localize to DDX6 granules: (i) RNase1 fused to RFP (RFP-RNase1) freely diffusing in the cytoplasm, and (ii) RNase1 fused to RFP N-terminally tagged with the first 99 nucleotides of the outer mitochondrial membrane protein TOMM20 (TOMM20-RFP-RNase1), resulting in the tethering of the reporter to the outside of mitochondria (Suppl. Fig. 5f)[47,48]. Equivalent reporters lacking the RNase1 tag were used as control. Both TOMM20-RFP reporters co-localized strongly with the mitochondrial protein cytochrome C (CYT C), spatially distinct to DDX6 granules (Suppl. Fig. 5g). The four RFP constructs were co-transfected with the GFP-DDX6 reporter to assess their effect on DDX6 granules (Suppl. Fig. 5h). None of the RFP reporters affected DDX6 granule size, number, or fluorescence intensity (Suppl. Fig. 5i-k). Together, these experiments indicate that RNase1 activity outside DDX6 granules does not account for the reduction in DDX6 granule size we observed by its direct tethering. This strongly suggests that RNA degradation at DDX6 granules drives their disassembly. Finally, we investigated whether the changes in DDX6 granule assembly dependent on neuronal activity (Fig. 2) would also require RNA supply. To this end, we transfected either our GFP-DDX6 or GFP-DDX6-RNase1 reporters in mature hippocampal neurons (Suppl. Fig. 5l) and inhibited neuronal activity overnight by combined application of CNQX, AP5, and TTX. We observed a clear increase in GFP-DDX6 granule size upon neuronal inhibition, validating our previous findings (Fig. 2) and confirming our overexpressed reporter phenocopied endogenous DDX6 (Fig. 5e; $F_{1,8} = 0.00045$, GFP reporter; $F_{1,8} = 0.0128$, neuronal inhibition). Interestingly, the smaller granules formed by GFP-DDX6-RNase1 did not significantly increase in size after neuronal inhibition (Fig. 5e). This strongly supports the notion that the assembly of DDX6 granules governed by neuronal inhibition requires RNA supply. Notably, the average number of GFP-DDX6 and GFP-DDX6-RNase1 granules remained largely unchanged in the cell body (Suppl. Fig. 5m).

To gain initial insight into whether DDX6 is involved in the regulation of synaptic function, we performed two pilot experiments with hippocampal neurons lacking DDX6. This assumption is based on the fact that DDX6 has been linked to translational regulation, an essential process for synapse formation[49,50]. Hippocampal neurons were transduced using a lentivirus expressing a short-hairpin RNA targeting *DDX6* mRNA (shDDX6) resulting in a ~ 40% reduction of DDX6

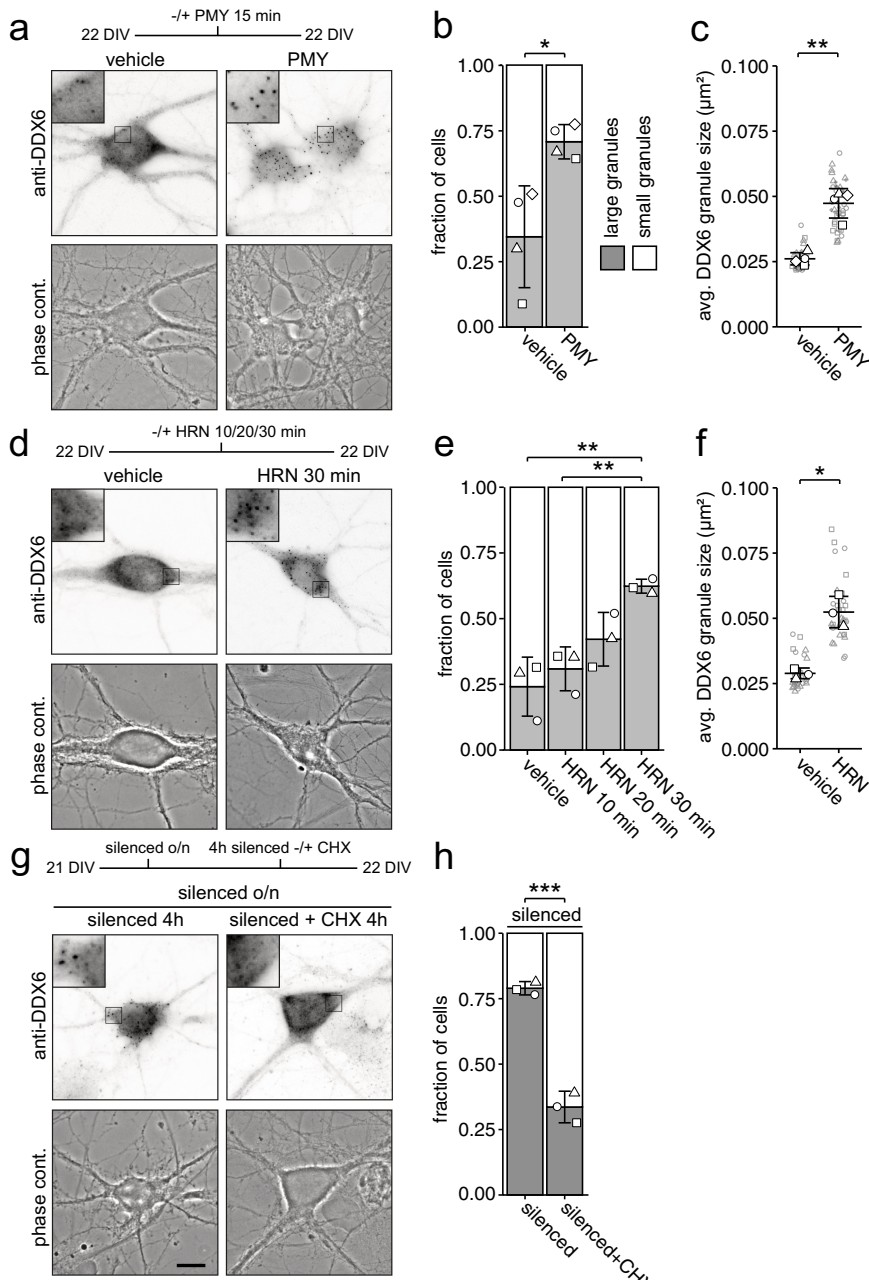

**Fig. 4 DDX6 granule assembly is facilitated by the availability of cytoplasmic non-translating mRNAs. a, d, g** Experimental outline and representative examples of DDX6 immunostainings and phase contrast pictures of 22 DIV hippocampal neurons in culture under 15 min vehicle or puromycin (25 μM) treated conditions (**a**), under 30 min vehicle (DMSO) or harringtonine (2 μg/mL) treated conditions (**d**), and under silenced (100 μM CNQX, 50 μM AP5, 1 μM TTX) conditions, followed by 4 h additional silencing or silencing + CHX (**g**). Abbreviations: PMY = puromycin, HRN = harringtonine, CHX = cycloheximide. Boxed regions in images are displayed as magnified insets. Scale bars 10 μm. **b, e, h** Bar plots displaying quantification of cell population by fraction of cells containing either large or small DDX6 granules as exemplified in (**a**), (**d**) and (**g**). Distinct dot symbols indicate biological replicates. At least 100 cells/condition/experiment were quantified. $n = 4$ (**b**) and $n = 3$ (**e, h**) biologically independent experiments). **c, f** Dot plots displaying average DDX6 granule size of individual cell bodies. Small gray symbols represent single cells while larger white symbols indicate the average of each replicate. Data represents mean of three or four independent neuronal cultures. $n = 4$ (**c**) and $n = 3$ (**f**) biologically independent experiments). Asterisks represent p-values obtained by two-sided Student's t-test (**b, c, f, h**) or Tukey's test post-hoc to one-way ANOVA analysis (**e**) (*$p < 0.05$, **$p < 0.01$, ***$p < 0.001$). $p = 0.0276$ (**b**), $p = 0.0022$ (**c**), $F_{3,8} = 0.00333$ (**e**), $p = 0.0096$ (**f**), $p = 0.00084$ (**h**).

protein and DDX6 granule size, mimicking the reduction in protein levels and granule size observed during neuronal maturation (Fig. 1a–c and Suppl. Fig. 1e, f). Firstly, down-regulation of DDX6 led to increased clustering of the postsynaptic marker Homer1 in dendrites of 15 DIV hippocampal neurons (Suppl. Fig. 5n, o). Secondly, Ca²⁺ live imaging showed an

increase in the frequency of fluorescent peaks in neurons where DDX6 was depleted (Suppl. Fig. 5p, q; $p = 0.043$, Suppl. Movie 3). Together, these preliminary data indicate that DDX6 is likely involved in the regulation of synaptic function. We propose that depletion of DDX6 may de-repress synaptically relevant mRNAs, resulting in de-regulated synapse formation and activity. Future

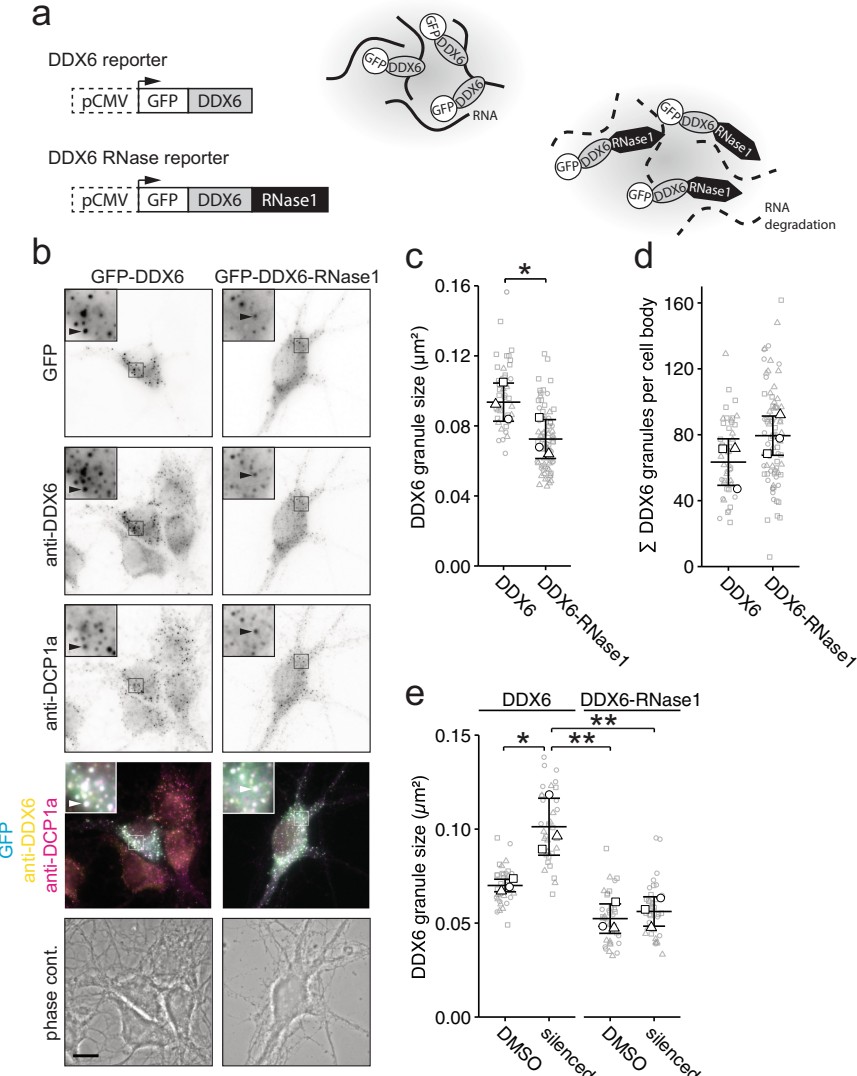

**Fig. 5 RNA degradation in DDX6 granules results in disassembly. a** Scheme of GFP-DDX6 and GFP-DDX6-RNase1 expression cassettes (right) and their use in an RNA-dependent assembly assay (left). **b** Representative examples of GFP fluorescence, anti-DDX6 and anti-DCP1a immunofluorescence, merge, and phase contrast pictures of 14 DIV hippocampal neurons in culture, transfected with either GFP-DDX6 or GFP-DDX6-RNase1 reporters. Boxed regions in images are displayed as magnified insets. Scale bars 10 μm. Arrowheads indicate overlap. **c**, **d**, **e** Dot plots displaying average DDX6 granule size (**c**, **e**) or DDX6 granule number (**d**) of individual cell bodies as exemplified in **b**, transfected with either GFP-DDX6 or GFP-DDX6-RNase1 reporters (**c**, **d**) and under vehicle treated or silenced conditions (**e**). Distinct dot symbols indicate biological replicates. Small gray symbols represent single cells while larger white symbols indicate the average of each replicate. Horizontal line and error bars represent mean of replicates and standard deviation ($n = 3$ biologically independent experiments). Asterisks represent *p*-values obtained by two-sided Student's *t*-test (**c**, **d**) or Tukey's test post-hoc to one-way ANOVA analysis (**e**) (*$p < 0.05$, **$p < 0.01$). $p = 0.0306$ (**c**), $F_{1,8} = 0.00045$, GFP reporter; $F_{1,8} = 0.0128$, neuronal inhibition (**e**).

studies will need to consolidate whether this is indeed the case and which relevant transcripts are regulated by DDX6 in neurons.

In summary, our results clearly underpin the importance of RNA for granule assembly in cells. Moreover, we show that the process of DDX6 complex formation is highly dynamic and relies on a fine-tuned balance between RNA supply and translational activity that is modified by synaptic activity and neuronal maturation.

## Discussion
Our work provides functional and mechanistic insights into how RNA granules might dynamically balance transcriptomic and translational changes and could respond to the demand for the translation of specific transcripts. Mammalian cells are able to shape protein expression by regulating the translatability of their

mRNAs[51,52]. In this respect, several independent mechanisms such as RNA transport[53], binding of translation regulators[6], or RNA degradation[54,55] have evolved to allow spatiotemporal sorting of mRNAs and the control of protein expression patterns. Recent work has shown that mRNAs facilitate condensation into RNA granules in order to regulate their expression in vitro[22,56,57]. This goes well in line with our finding that DDX6 granule assembly is affected by processes regulating the amount and availability of cytosolic RNA, such as translation activity and synaptic transmission. This mechanism involving the DEAD box RNA helicase DDX6 seems to be evolutionarily conserved, as a parallel study in *Drosophila*[58] found that granule formation of the DDX6 homolog Me31B during aging affects the translatability of mRNAs in the fly brain. Together, this indicates that DDX6 granules regulate transcripts during neuronal maturation and synaptic transmission by the assembly and disassembly of RNA

granules. Though DDX6 and P-bodies have been extensively studied as translational repressors and regulators of mRNA and DDX6 is a core component of P-bodies[11,46], it remains to be seen whether DDX6 function itself directly affects RNA-dependent granule assembly and disassembly. Moreover, we do not yet know whether DDX6 binds free mRNA in a non-specific manner, or rather interacts with specific targets during the assembly processes observed in our study. Future studies will have to unravel how this DEAD box RNA helicase conveys RNP assembly in neurons and how this process is regulated by maturation and synaptic activity. Indeed, DEAD-box ATPases have been identified as regulators of RNA granule formation[25,59]. Interestingly, eIF4A, another RNA helicase in mammals, has been implicated in the regulation of RNA-RNA interaction dependent assembly, and limits the formation of RNA condensates[15]. These findings are in line with an in vitro study showing that RNA can condensate through intermolecular RNA-RNA interactions[56].

In addition to RNA availability, we have recently identified RNA secondary structures as another important feature in the process of RNA granule assembly. Here, specific RNA hairpins bound by Stau2 regulate granule assembly, translatability, and dendritic transport of target RNAs[36]. Consequently, RNAs can drive granule assembly not only through sequences but also through secondary structures in vivo. Here, we show that the condensation of DDX6 granules requires Stau2 dependent RNA regulation during synaptic inhibition. This is most likely due to the involvement of Stau2 in the transport and redistribution of mRNAs. However, we cannot exclude that the translational regulation of specific target mRNAs by Stau2 may contribute to this process as well. Together, our data clearly support a working model in which RNA supply and neuronal activity critically determine DDX6 granule formation in cells. In turn, depletion of DDX6 results in altered synaptic activity. This mechanism of RNA granule assembly would allow cells to cope with different physiological conditions and to dynamically balance the demand for the translation of specific transcripts, via the reversible assembly of RNA granules. Therefore, the study of RNA granule formation is important for our understanding of physiological processes, such as synaptic plasticity in neurons, which heavily depends on mRNA regulation[60,61]. Moreover, unraveling the underlying molecular principles of RNA granule assembly is a prerequisite for our understanding of the physiological impact of foreign RNA exposed in a cell, *e.g.* complex RNA viruses such as SARS-CoV-2 or the efficiency of mRNA vaccines[62,63].

## Methods

**Neuronal cell culture and transfection**. Primary rat hippocampal and cortical neuronal cell cultures were generated as previously described[64]. In short, hippocampi of embryonic day 17 (E17) embryos of timed pregnant Sprague-Dawley rats (Charles River Laboratories) were isolated, cells dissociated, and plated on Poly-L-Lysine coated coverslips and cultured in NMEM + B27 medium (Invitrogen). Experiments were performed with cultured neurons between 8–29 days in vitro (DIV). Neurons were transiently transfected by calcium phosphate co-precipitation as previously described[65]. A total amount of 3 µg DNA was used for single or co-transfected plasmids. All animals were used according to the German Welfare for Experimental Animals (LMU-Munich, *Regierung von Oberbayern*).

**Cryosections**. Eight day postnatal and 10 month old male black 6 mice (C57B/6 J, Jackson Laboratory) were anesthetized and intracardially perfused with PBS (Invitrogen) followed by 4% paraformaldehyde (PFA) and processed as published[66,67]. In brief, brains were removed, postfixed in 4% PFA overnight, washed 3x with PBS and placed in cold 30% sucrose in PBS till they sunk down (~2 days). Samples were embedded in OCT (Tissue-Tek) and cryopreserved. Sagittal cryosections (40 µm thick) were permeabilized with PBS-0.1% Triton X-100 (PBT) and then blocked with 1% BSA in PBT. Primary polyclonal goat anti-DDX6 (Abnova) antibody was incubated o/n at 4 °C. Secondary calf anti-goat Alexa488 or Alexa555 conjugated antibodies (Dianova) were incubated for 2 h at RT. Slides were mounted with Aqua-Poly/Mount (Polysciences).

**Plasmids**. The DDX6, RNase1, and the first 99 nucleotides of TOMM20 sequences were obtained by PCR amplification from rat cDNA without a functional stop codon. The pEGFP-DDX6 plasmid was generated by placing the DDX6 sequence in frame with GFP in the pEGFP-C1 vector (Clontech). The pEGFP-DDX6-RNase1 plasmid was obtained by placing the RNase1 sequence in frame into the pEGFP-DDX6 vector. A pTOMM20(1-99)-TagRFP plasmid was generated by placing the pTOMM20(1-99) sequence in frame with TagRFP in the pTagRFP-C vector (Evrogen). pTagRFP-RNase1 and pTOMM20(1-99)-TagRFP-RNase1 plasmids were obtained by cloning the RNase1 sequence in frame with pTagRFP-C (Evrogen) and pTOMM20(1-99)-TagRFP vectors. The pGP-CMV-GCaMP6s plasmid for Ca$^{2+}$ imaging was obtained from Addgene (#40753)[68].

**Chemical treatments**. To inhibit neuronal activity, cells were treated as published[30]. In brief, cells were incubated with 100 µM 6-cyano-7-nitroquinoxaline-2,3-dione (CNQX; Sigma, #C127), 50 µM 2-amino-5-phosponopentanoic acid (AP5; Sigma, #A8054) and 1 µM tetrodotoxin (TTX; Abcam, #ab120055) in NMEM + B27 medium overnight at 37 °C, unless otherwise stated. Vehicle treated cells were incubated with an equivalent amount of DMSO. Wash off experiments were performed by a short wash in pre-warmed HBSS and subsequent 15 min recovery in NMEM + B27 medium at 37 °C. Stimulation by NMDA was done by a quick wash with pre-warmed PBS and 15 min incubation with 100 µM NMDA in NMEM + B27. Cycloheximide (CHX, 7 µM, Roth) was incubated for 4 h, puromycin (PMY, 25 µM, Sigma-Aldrich) was incubated for 15 minutes and harringtonine (HRN, 2 µg/mL, Biomol) was incubated for 10, 20 or 30 minutes in NMEM + B27 medium before fixation.

**Lentivirus production**. Control shNTC, shStau2-2[34] and shDDX6 lentiviral particles were obtained from HEK293T cells (ATCC CRL-3216, authentication by ATCC) co-transfected with the plasmids psPAX2, pcDNA3.1-VSV-G and either pFu3a-H1-shNTC-pCaMKIIα-tagBFP, pFu3a-H1-sh-Stau2-2-pCaMKIIα-tagBFP, pFu3a-H1-shNTC-pCaMKIIα-tagRFP or pFu3a-H1-shDDX6-pCaMKIIα-tagRFP, respectively, using calcium phosphate co-precipitation. The sequence of shDDX6 was GATCCCCTTTATCTGGTAGGGATATCTTCAAGAGAGA-TATCCCTACCAGATAAATTTTTA. Supernatants were filtered (0.45 µm RVDF Millex-HV; Millipore), concentrated by ultracentrifugation (23,000 rpm, 140 min, SW 32 Ti rotor; Beckman Coulter), and resuspended in Opti-MEM™ (Life Technologies)[69]. Hippocampal neurons were transduced for 4 days before fixation.

**Immunostaining**. Neurons were fixed for 10 min with 4% PFA and immunostained as described[34]. The following primary antibodies were used: polyclonal rabbit anti-Rck 1:500 (MBL), polyclonal goat anti-DDX6 1:500 (Abnova), polyclonal mouse anti-DCP1a 1:500 (Abnova), polyclonal rabbit anti-G3BP1 1:500 (Proteintech), monoclonal mouse anti-Stau2 1:500[18], monoclonal mouse anti-CYT C 1:200 (Biolegend, clone 6H2.B4) and mouse monoclonal anti-Homer1 1:500 (Synaptic Systems); and detected by the following secondary antibodies: donkey anti-rabbit or donkey anti-mouse AlexaFluor488, AlexaFluor555 or AlexaFluor647 conjugated antibodies 1:1000 (Life Technologies).

**Microscopy**. Imaging of sagittal brain sections was performed at the core facility bioimaging of the Biomedical Center with an inverted Leica SP8 microscope, equipped with a Plan-Apochromat 63x objective, a laser for 552 nm excitation, hybrid photo detectors (HyDs), and a convectional photomultiplier tube (PTM; Hamamatsu R 9624). Image pixel size was 361 nm. Z-stacks were acquired at 700 nm interval. Imaging of fixed cells was performed on a Zeiss Z1 Axio Observer microscope, including a Plan-Apochromat 63x objective, a COLIBRI.2 LED light source and the Axiocam 506 mono camera. Live cell imaging was performed on a Zeiss Cell Observer spinning disk system, consisting of a Zeiss Z1 Axio Observer microscope including a Plan-Apochromat 63x objective, a Yokogawa CSU-X1 spinning disk unit with 4 laser lines (405 nm 20 mW; 488 nm 50 mW, 561 nm 75 mW and 638 nm 75 mW) and an Evolve 512 Delta EMCCD Camera. A custom made EMBL environmental chamber (EMBLEM) was used for temperature control. Hippocampal neurons were imaged at 36.5 °C in HBSS (Life Technologies) supplemented with 20 mM HEPES buffer pH=7.3 (Sigma Aldrich). Time-lapse images were acquired at an approximate frame interval of either 30 or 0.5 sec. Z-stacks at 1 µm intervals spanning the entire cell body were acquired per time-point for live imaging of NMDA and wash-off treatments, and subsequently subjected to maximum intensity projection. Cells were selected for proper expression of plasmid as well as for cell morphology and cell viability.

**Image data analysis**. Assessment of neuronal population with large or small granules (categorized as exemplified in Fig. 1a by 8 DIV vs. 29 DIV, and Suppl. Fig. 1a) was done by manually scoring >100 cells/condition/experiment. Detailed quantification of granule size and number was performed using the thresholding function of the Arivis Vision 4D software. Quantification of granular vs. cytoplasmic fluorescence was performed with a custom written ImageJ[70] script, available upon request. In short, images were gamma adjusted, a Laplacian filter was applied and images were thresholded using the automatic MaxEntropy method, to generate a detection mask for the granular compartment. A second detection mask for an approximation of cytoplasmic intensity outside granules was

generated by dilating the first mask by 2 pixels and subtracting the first mask, leaving only the dilated part outside granules. Both masks were used to measure the mean fluorescence in the original unprocessed images.

For deconvolution, z-stacks were acquired at a distance of 0.26 μm. Z-stacks were subjected to deconvolution using the constrained iterative quantitative restoration method of the Zeiss ZEN software deconvolution module.

**Statistical analysis**. The R[71–74] or GraphPad Prism 5 software were used for data processing, plotting, and statistical analysis. Figures represent mean ± standard deviation of at least 3 independent experiments, unless otherwise stated. Asterisks represent $p$-values obtained by either Student's $t$-test or Tukey's test post-hock to one-way ANOVA analysis ($*p < 0.05$, $**p < 0.01$, $***p < 0.001$), as indicated. The subscript of $F$ values denotes the degrees of freedom. All statistical tests were performed on individual biological replicates.

**Polysome profiling**. Polysome profiling was performed as described[75]. In brief, 4 million cortical neurons were treated either with 100 μg/mL cycloheximide (CHX) or, for ribosome runoff experiments, with 2 μg/mL harringtonine (HRN) for 10 minutes at 37 °C. Cells were washed with prewarmed HBSS supplemented with CHX or HRN and lysed in polysome lysis buffer (150 mM NaCl, 5 mM $MgCl_2$, 10 mM Tris-HCl pH 7.4, 1% NP-40, 1% (w/v) sodium deoxycholate supplemented with 100 μg/mL CHX and 2 mM dithiothreitol (DTT). Lysates were spun at 13,000 x $g$ for 5 min at 4 °C. Supernatants were loaded onto a sucrose density gradient (18% (w/v) to 50% (w/v) sucrose in 100 mM KCl, 5 mM $MgCl_2$, 20 mM Hepes pH 7.4). Gradients were centrifuged at 35,000 rpm (SW55Ti, Beckman) for 1.5 h at 4 °C. Gradients were fractionated in 10×500 μL fractions using an automated fractionator (Piston Fractionator, Biocomp). Proteins were extracted using methanol/chloroform extraction[76]. Individual fractions were subjected to Western blot analysis using polyclonal rabbit anti-Rck 1:1000 (MBL) or polyclonal rabbit anti-RPL7A 1:1000 (Abcam) primary and the donkey anti-rabbit IRDye 680RD conjugated 1:10,000 (LI-COR) secondary antibodies.

**Differential centrifugation**. Differential centrifugation was performed as described[77,78]. In brief, 16 million cortical neurons were lysed in homogenizing buffer (HB; 150 mM KCl, 50 mM Hepes pH 7.4, 1x complete protease inhibitor [Roche], 5 μL Ribolock [ThermoFisher] per 10 mL HB) on ice. Homogenate was spun at 16,000 × $g$ for 10 min at 4 °C (S16, P16). Supernatant S16 was then spun at 100,000 × $g$ for 20 min at 4 °C (S100, P100). When required, samples were treated with RNase1 (Ambion) prior to centrifugation. P100 pellets were volume-even resuspended in RIPA buffer (150 mM NaCl, 50 mM Tris-HCl pH 8, 0.5% (w/v) sodium deoxycholate, 1 vol% NP-40, 0.1% (w/v) SDS, 1× complete protease inhibitor, Roche) at 37 °C. All fractions (S16, P16, S100, P100) were methanol/chloroform extracted as described[76]. Samples were analyzed by Western blot, using polyclonal goat anti-DDX6 1:1,000 (Abnova) and mouse anti-Gephyrin 1:1,000 (Synaptic Systems) primary and donkey anti-goat IRDye 680RD conjugated 1:10,000 and donkey anti-mouse IRDye 800CW conjugated 1:10,000 (LI-COR) secondary antibodies.

**GFP-immunoprecipitation/pulldown and in vitro RNA digest**. 20 μL of slurry protein G sepharose beads (GE Healthcare) were washed with IP-buffer and coupled with 100 μg of monoclonal anti-GFP antibody diluted in IP-buffer overnight at 4 °C. Beads were blocked with 1.25 mg/mL BSA in IP-buffer overnight at 4 °C. pEGFP-DDX6 and pEGFP-DDX6-RNase1 plasmids were transiently transfected in HEK 293 T cells. For IP, cells were lysed in IP-buffer (50 mM Tris-HCl pH 7.4, 150 mM NaCl, 4 mM EDTA pH 8.0, 1% Triton-X100, 0.1% SDS, 1× complete protease inhibitor [Roche]) and homogenized using ultra sonication. Lysate was spun at 16,000 × $g$ for 10 min at 4 °C. The resulting supernatant (S16) was added to the beads and incubated for 4 hours at 4 °C. Beads were washed with IP-buffer and then either used for RNA digestion or protein analysis. For RNA digestion, 4,7 μg of total rat brain RNA was incubated with the beads at 22 °C for 30 min. RNA samples were subsequently loaded on a 1% agarose gel. For Western blot analysis, beads were boiled in SDS-sample buffer and analyzed by immunoblotting using the mouse anti-GFP 1:500 (self-made, kind gift by Angelika Noegel, Köln) as primary antibody and donkey anti-mouse IRDye 800CW 1:10000 (LI-COR) as secondary antibody.

**Western blotting**. Western blot analysis was performed as described[75]. In brief, protein samples were transferred on a nitrocellulose membrane and blocked in 2% (w/v) BSA. Proteins were detected using rabbit anti-Rck 1:1000 (MBL), goat anti-DDX6 1:1,000 (Abnova), mouse anti-ß-actin 1:5000 (Sigma-Aldrich), rabbit anti-RPL7A 1:1000 (Abcam), mouse anti-Gephyrin 1:1000 (Synaptic Systems) and mouse anti-GFP 1:500 (self-made, kind gift by Angelika Noegel, Köln) primary antibodies and donkey anti-rabbit IRDye 680RD conjugated 1:10,000, donkey anti-mouse IRDye 800CW conjugated 1:10,000 and donkey anti-goat IRDye 680RD conjugated 1:10,000 (LI-COR) secondary antibodies. Primary antibody binding was detected using the LI-COR Odyssey IR scanner.

**Reporting summary**. Further information on research design is available in the Nature Research Reporting Summary linked to this article.

## Data availability
The data supporting the findings of this study are available from the corresponding authors upon reasonable request. Uncropped blots and gels are provided in Supplementary Information. Source data for the figures and supplementary figures have been deposited at: https://osf.io/c7s6p/

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

## Acknowledgements

We thank Sabine Thomas and Nicole Kerckhoff for primary neuron culture preparation; Manja Thorwirth for perfusion of mice; Renate Dombi for cloning; Daniela Rieger for antibody purification; Sandra Fernandez-Moya and Janina Ehses for lentiviral production; Angelika Noegel for GFP monoclonal antibody clone; Sabine Thomas, Marco Tolino and Saskia Hutten for initial experiments; and Edouard Bertrand, Florence Besse, Kavya-Vinayan Pushpalatha, Sandra Fernández-Moya, Janina Ehses and Barbara Nitz for advice and critical comments on the manuscript. This work was supported by grants from the DFG (Großgeräteantrag INST86/1581-1FUGG, SPP1738, FOR2333, SFB870), the FWF (I 590-B09, F4314-B09 SFB RNA-seq) (all to MAK), the DFG (project number 413985647 to MH), the Friedrich-Baur-Stiftung (to IS and independently to MH) and WiFoMed (to KEB).

## Author contributions

M.A.K. conceived the project. K.E.B., N.B., R.S., C.I., I.S., and M.H. performed experiments. All authors were involved in data analysis. K.E.B., R.S., and M.A.K. wrote the manuscript. All authors approved the manuscript before submission.

## Funding

## Competing interests

The authors declare no competing interests.
