## [Peer Review File · Nature Communications]

Title: RNA supply drives physiological granule assembly in neuronsREVIEWER COMMENTS

Reviewer #1 (Remarks to the Author):

In this manuscript, Bauer and colleagues describe how the RNA helicase, DDX6, is involved in granule assembly within neurons. The authors find DDX6 granules decrease in size during in vitro culturing of neurons, which they also report in tissue. The authors use these findings as a starting point for the experiments that follow.

To define the mechanism of DDX6 granule assembly and disassembly, the authors provide strong evidence that large DDX6 granules are augmented by neuronal silencing (CNQX/AP5/TTX treatment). Their work shows Staufen 2 has an important role in DDX6 granule formation using a knockdown approach after neuronal silencing. They suggest that DDX6 granules are transiently interacting with polysomes as supported by their translation inhibition and polysome studies. Lastly, they propose RNA are involved in DDX6 granule assembly within P-bodies. It is clear from their work that DDX6 granule size was decreased using the DDX6-RNase reporter. However, the specificity of the DDX6-RNase reporter is unclear, and clarification of this specificity would strengthen the authors findings. Overall, this work is interesting and provides important mechanistic insights into protein granule assembly and disassembly.

Major comments:

- The authors provide evidence for the mechanism of large DDX6 granule assembly throughout the paper. However, there are several figures where the characterization of the small and large granules is omitted. The division of granules based on size was ambiguous and there was no clear cut-off mentioned. To clarify the cut-off for small vs. large granules, please provide the total number, mean fluorescence intensity, and average number of small vs. large granules (Figure 1C-E and Figure 5C-F).
- Figure 5. The RNase reporter is active throughout the cell. Therefore, it is likely reducing total RNA content outside of P-bodies. The authors suggest that RNA availability is affecting the granule assembly within P-bodies. Could the authors show that the RNA within P-bodies is also affected?

Minor comments:

- Figure 1B. The fraction of cells for 22 DIV has a negative error bar, which is not correct as a fraction cannot be negative. Perhaps the authors can either increase sample size or set SEM lower boundary to "0".
- The authors always report large and small granules as a "fraction of cells." Could the authors clarify what this means in the context of small vs. large granules? For example, do the authors mean the fraction of cells with large granules or the fraction of large granules per cell?
- Similarly, the fraction of cells has large variability. Could the authors please report the absolute number of granules (large and small) to show whether this variability is due to low numbers for a given cell?

- Figure 5C-F. Could the authors clarify whether the DDX6 granule quantifications based on the colocalization of all staining (GFP, DDX6 and DCP1a)?
- Figure 1A, 2A, 3A and 4. The emphasis should be on the granules not on cell morphology. For figures with phase contrast images, either remove the phase contrast or place the inserts (zoomed in images) before the phase contrast images. Also, indicate/label examples of small and large granules in the zoomed insets.
- Figure 3. Include quantification of granule size and mean fluorescence. Also, include co-immunofluorescence of Stau2 and DDX6 to determine whether they are colocalized within the granules.

Reviewer #2 (Remarks to the Author):

In this manuscript Bauer et al assess DDX6 granules formation during hippocampal neuron maturation, and investigate the role of RNA in this granule formation. First, the authors use hippocampal neurons to make the interesting observation that RNA granule size decreases as neurons mature in culture, and further observe similar correlations in vivo. The authors then demonstrate that inhibition of synaptic activity results in larger granules, whereas activation has a reciprocal impact. They use knockdown of Stau2 RNA binding protein to show that granule assembly following neuronal inhibition depends upon Stau2. Manipulating translation with inhibitors which release RNA also impacts granule size, as does mis-expressing DDX6-RNAase. These two experiments suggest that depleting RNA disrupts DDX6 granule formation.

The study presented by Bauer et al has some interesting findings, including the observation that RNA granules change during neuronal maturation and in response to synaptic activity. However there are several concerns with the paper-as it feels somewhat preliminary, the flow of the paper somewhat disjointed and the motivation for specific experiments is often unclear. Furthermore, while the experiments are generally well rigorous there are some missing controls that affect the impact.

Main concerns:

- 1) Overall the results in this paper are quite preliminary. While the observation that RNA promotes granule formation is interesting, this is not a new concept. Unfortunately the impact of their findings alone are limited as they are very correlative. What is the functional impact of changing DDX6 granule size or showing that it depends upon RNA? Does granule size affect neuronal activation or synaptic function? Extending their study to other granules may also increase the impact of this study as would showing that it was functionally important for the cells. It sounds like one of the co-submitted papers may do this, and the authors may reconsider combining data as appropriate.
- 2) The quantification of experiments and description could be improved. First, there were insufficient details provided for how the granules were classified as large or small. Please provide exact cut-offs and rationale for these cutoff. (For example in Figure 1C there are about 12% of granules which are large

(fig 1B) but how does that equate to data in Fig 1C? Is it the few outliers? How was this selected?)

Second, the data in Figure 1F reflect a shift to more cytoplasmic signal of DDX6. To what extent does this reflect more cell area (differences in size of older neurons?)

3) A major point of the paper is that RNA availability affects granule size, but this observation is limited to analyses at one stage (either with translational inhibitors or with GFP-DDX6-RNaseI). Are the changes in granule size seen during neuronal maturation (Fig 1) and neuronal activity (Fig 2) also driven by RNA? The role of RNA in granule formation in these contexts could be further fleshed out to increase the impact of their findings.

Likewise, the observation that silencing synaptic activity leads to larger granules is interesting, especially since this is more similar to an immature neuron. This begs the question of how granule formation changes with maturation—are the granules less mobile when they are larger or in response to silencing synaptic activity? Fleshing their observations out on this front, with live imaging to assess the mobility of granules of different sizes, or understand if granules coalesce with maturation or grow larger from independent granules, could also be interesting.

4) Given the difference in granule size in response to changes in synaptic activity, it would be important to look at granules sub-cellularly, at the synapse or in dendrites specifically (Fig 2). Likewise, as Stau2 is important in mRNA localization and particularly to DDX6 granules, it would be valuable to assess how Stau2 KD affects granules at localized compartments (ie: soma versus synapse and/or dendrites).

5) The motivation for manipulating Stau2 was unclear. Perhaps the co-submitted paper would help clarify, but in the context of this manuscript, it remains unclear.

6) The translation inhibitor experiments could be better developed. First, it is unclear why they only use CHX under silenced conditions. The comparison with PMY and HRN at baseline would be important for interpreting their findings. Second, translation inhibitors will do a lot more to the cell than simply release RNA. How can the authors rule out that observed differences are not due to some other impact of inhibitors?

7) The tethering of RNaseI to GFP-DDX6 affects total RNA availability (Fig. 5 and S5), which then influences DDX6 granule size. This assay needs some additional controls. How can the authors be confident that this tag does not affect DDX6 localization in the cell?—the pattern of GFP looks different and this needs to be assessed. It would be valuable to also include an experiment where RNaseI is tethered to a non-RBP or an RBP found in a distinct granule (ie: stress granules; perhaps G3BP1) to show specificity of this finding.

Minor concerns:

1) The authors may consider using super-plots to represent their data (Lord, SJ, et al, JCB 2020).

2) It is unclear whether the N for the statistics is # of cells or independent experiments. Using the number of cells quantified for statistical tests is inappropriate.

3) In Fig 2C, the baseline fraction of cells with large granules is different than in Fig 1 B (40% v 25%)

4) Need references for statement that synaptic activity determines localization and translatability of mRNAs (figure 3)

5) They show that Stau2 is required for formation of large granules after silencing activity. Is the reverse

true? Does Stau2 overcome formation of small granules following activation?

6) Use of word *in vivo* needs to be tempered in the conclusions, as all experiments besides immunofluorescence in Fig 1 are *in vitro* experiments

Reviewer #3 (Remarks to the Author):

The manuscript by Bauer et al. examines how the availability of RNA shapes DDX6 granule condensation using an *in vivo* system of maturing neurons in culture. The authors begin by describing a gradual decondensation of DDX6 granules upon neuronal maturation, both in cultured neurons and mice brain sections. They then show that DDX6 granules can assemble in response to synaptic inhibition in a reversible manner. This inhibition-induced assembly is sensitive to Stau2 knockdown (with the knockdown alone not affecting DDX6 granule size). The authors then compare three translation inhibitors that differentially affect the pool of ribosome-associated RNAs and provide evidence that the availability of ribosome-free cytoplasmic RNA is a determinant of granule condensation. Finally, the authors use a DDX6-RNase1 fusion to locally deplete RNA from granules and observe a reduction in granule size further demonstrating the importance of RNA in granule condensation.

Investigating the relationship between DDX6 granule dynamics and RNA availability in a physiological context is interesting and adds new insight to the field of granule biology. This manuscript is well written, concise, and easy to follow. Experiments are nicely controlled and clearly presented. The paper's conclusions are overall properly supported by the data.

I have one major comment that needs to be addressed before recommending the manuscript for publication:

To directly demonstrate the effects of RNA depletion in granules, the authors induce localized RNA degradation through a GFP-DDX6-RNase1 fusion. My concern here is that a fraction of GFP-DDX6-RNase1 does not localize in granules (as demonstrated by the diffuse cytoplasmic GFP and IF signals in Fig. 5B for instance, and the mean fluorescence granular/cytoplasmic ratio in Fig. 1E). This cytosolic non-granule fraction of GFP-DDX6-RNase1 will therefore deplete non-granule RNA. It will thus improve this manuscript to prove that this system preferentially degrades RNA in granules. To demonstrate this, I suggest (i) adding smFISH experiments quantifying a granule-enriched and a non-enriched transcript with GFP-DDX6 or GFP-DDX6-RNase1 expressed, (ii) and/or comparing the measured granule parameters with an additional construct containing RNase but not fused to DDX6 or any granule marker. These experiments will consolidate the notion that RNAs, specifically in granules, are a determinant of condensation.

Minor comment:

To provide a more complete view on the role of RNA availability in granule formation, please mention studies performed using artificial granules such as PMID: 31324804.

Response to Reviews

We sincerely appreciate the reviewers' assessment of our previous submission and their comments on improving the manuscript. We thank all reviewers for the positive assessment of our work, *e.g.* "overall, this work is interesting and provides important mechanistic insights into protein granule assembly and disassembly", "the experiments are generally well rigorous" and the work "adds new insight to the field of granule biology", the "manuscript is well written, concise, and easy to follow", "experiments are nicely controlled and clearly presented" and finally, "the paper's conclusions are overall properly supported by the data."

Based on their comments, we have made substantial revisions to the manuscript and incorporated a series of new figures both in the main manuscript as well as in the Supplements. We therefore hope that the reviewers as well as the editor will agree that these substantially strengthen the manuscript and now provide independent lines of evidence supporting our working model. As there are a couple of concerns that were raised by more than one reviewer, we would like to address those upfront, before we present a point-by-point response below with the reviewers' comments italicized and our response in blue text.

A. Specificity of RNase1 activity during DDX6-RNase 1 expression

All reviewers raised questions concerning the specificity of our GFP-DDX6-RNase1 reporter, specifically whether RNase1 activity outside DDX6 granules would elicit the same reduction in DDX6 granule size as by direct tethering to granules (original **Figure 5**). Reviewers 2 and 3 specifically suggested to either use RNase1 not fused to any marker or fused to a non-RNA-binding protein. Therefore, we designed a new set of reporters based on RFP as a transfection marker (revised **Supplementary Figure 5F, see relevant panels below**). To investigate the effects of untethered RNase1, freely diffusing in the cytoplasm, we transfected either RFP fused to RNase1 (RFP-RNase1) or RFP alone. Moreover, to investigate the effects of RNase1 activity in locations different to DDX6 granules, we designed RFP reporters tethered to the outside of mitochondria. To this end, we linked the first 33 amino acids of the outer mitochondrial membrane protein TOMM20 to the N-terminus of RFP, resulting in the tethering to mitochondria, with the RFP facing the cytoplasm, and either included or omitted RNase1 C-terminally. Both TOMM20-RFP reporters co-localized strongly with the mitochondrial protein cytochrome C (CYT C), spatially distinct to DDX6 granules (**Supplementary Figure 5G, see below**). These four RFP constructs were co-transfected with the GFP-DDX6 reporter to assess their effect on DDX6 granules (**Supplementary Figure 5H, see below**). Detailed particle analysis revealed that none of the RFP reporters affected DDX6 granule size, number or fluorescence intensity (**Supplementary Figure 5I-K, see below**). Together, these new experiments indicate that RNase1 activity outside DDX6 granules, *i.e.* either diffuse or localized to mitochondria, does not account for the reduction in DDX6 granule size we observe by direct tethering in our system. This strongly suggests that RNA degradation at DDX6 granules drives their disassembly. We strongly agree with the reviewers that this represents an important control that we now include in the revised **Supplementary Fig. 5**. In fact, due to the high dynamicity and turnover of granule components such as RNA or DDX6, it is very plausible that a strong global reduction in RNA levels would eventually affect DDX6 granules. Moreover, the expression of the DDX6-RNase1 reporter showed a clear but modest reduction of mRNA levels in neurons by polyA-FISH (**Figure R2A-B, see below, for the reviewers only**). Therefore, it is likely that our RNase1 reporters display intracellular RNase activity that is not strong enough to elicit global responses, but sufficient to affect local compartments, when specifically targeted therein. A second independent line of evidence

supporting this view is that globally restricting mRNA in polysomes via the application of cycloheximide (CHX) reduced the amount of available mRNA and led to DDX6 granule disassembly (**Figure 4**). Conversely, increasing the cellular fraction of mRNA not associated with ribosomes by puromycin (PMY) or harringtonine (HRN) led to increased DDX6 assembly.

Selective panels F-K from **Supplementary Figure 5F-K (complete Figure in Supplements)**: RNase1 activity outside DDX6 granules does not affect granule assembly. (**F**) Scheme of RFP, RFP-RNase1, TOMM20-RFP, TOMM20-RFP-RNase1 and GFP-DDX6 constructs. (**G**) Representative examples of RFP fluorescence, anti-cytochrome C immunofluorescence and merged pictures of 17 DIV hippocampal neurons, transfected with TOMM20-RFP or TOMM20-RFP-RNase1 reporters, respectively. (**H**) Representative examples of GFP and RFP fluorescence in 17 DIV hippocampal neurons, co-transfected with GFP-DDX6 and RFP reporters, respectively. (**I, J, K**) Dot plots displaying average DDX6 granule size (I), granule number per cell body (J) and average fluorescence intensity (K). Distinct dot symbols indicate biological replicates. Small grey symbols represent single cells while larger white symbols indicate the average of each replicate. Horizontal line and error bars represent mean of replicates and standard deviation. Tukey's test post-hoc to one-way ANOVA analysis did not reveal any statistical significance.

B. DDX6 granule assembly upon synaptic inhibition is dependent on RNA

One very interesting and relevant question by reviewer 2 was whether the changes in granule assembly seen during neuronal maturation (**Fig. 1**) or neuronal activity (**Fig. 2**) would also be dependent on RNA. Showing that this is the case would further link our initial findings that neuronal function affects DDX6 granule assembly with our findings that RNA supply facilitates assembly. To address this question, we transfected our GFP-DDX6 and GFP-DDX6-RNase1 reporters in mature hippocampal neurons (revised **Suppl. Fig. 5L, shown on page 2**). After 24 hours of expression, we additionally inhibited neuronal activity over night by combined application of CNQX, AP5 and TTX (as performed in **Fig. 2**). Detailed particle analysis revealed a clear increase in GFP-DDX6 granule size upon neuronal inhibition, validating the findings of **Fig. 2** and confirming our overexpressed reporter phenocopies endogenous DDX6 (revised **Fig. 5E**). Interestingly, the smaller granules formed by GFP-DDX6-RNase1 did not significantly increase in size after neuronal inhibition. This strongly supports the notion that the assembly of DDX6 granules governed by neuronal function requires RNA supply. As we consider this a new and essential major finding, we included this experiment in **Fig. 5** and **Suppl. Fig. 5** of the revised manuscript. Considering the fact that DDX6 acts as a translational regulator, and such granules are thought to contain translationally repressed mRNAs (Dahm & Kiebler, 2005, Nature, PMID: 16306974; Fritzsche et al., 2013, Cell Reports, PMID: 24360960), our new data indicates that mRNA may be specifically targeted for translational repression in DDX6 granules dependent on neuronal activity. Together, it supports a physiological role for DDX6 granule assembly. Additionally, the formation and maintenance of synapses upon neuronal activation, during processes such as learning or memory formation, requires *de novo* protein synthesis (Sutton & Schuman, 2006, Cell, DOI: <https://doi.org/10.1016/j.cell.2006.09.014>). It is tempting to speculate that DDX6 granule disassembly upon neuronal stimulation (as seen in **Fig. 2E**) may release previously repressed mRNAs, making them available for subsequent translation. We believe that this new line of evidence raises the impact of our findings and elegantly bridges neuronal function, RNA supply and DDX6 granule assembly.

C. Reanalysis of data

Reviewers 1 and 2 requested a more detailed and clearer description of granule parameters for some of the experiments. To provide detailed insight, we reanalyzed the data of **Figs. 2, 3 and 4** by single particle analysis and included the data in the respective revised figures. Though a substantial amount of work, we believe this in depth look into the data is indeed very valuable in further supporting our initial findings. Due to the fact that NMDA and cycloheximide treatments resulted in a very diffuse DDX6 pattern, we were not able to perform single particle analysis on these datasets with the resolution of our conventional widefield microscope. However, we believe the respective effects are clear to see in the manual quantification of cell population and the representative images. In addition, reviewer 2 advised us to plot our data as super-plots. We find this representation very attractive and adapted our plots accordingly. Moreover, we replotted all graphs to clearly indicate the individual data points of each biological replicate. We agree that this representation provides more detailed and meaningful insight into the data.

Point-by-point response to the reviewers' comments

Reviewer #1 (Remarks to the Author):

In this manuscript, Bauer and colleagues describe how the RNA helicase, DDX6, is involved in granule assembly within neurons. The authors find DDX6 granules decrease in size during in vitro culturing of neurons, which they also report in tissue. The authors use these findings as a starting point for the experiments that follow.

To define the mechanism of DDX6 granule assembly and disassembly, the authors provide strong evidence that large DDX6 granules are augmented by neuronal silencing (CNQX/AP5/TTX treatment). Their work shows Staufen 2 has an important role in DDX6 granule formation using a knockdown approach after neuronal silencing. They suggest that DDX6 granules are transiently interacting with polysomes as supported by their translation inhibition and polysome studies. Lastly, they propose RNA are involved in DDX6 granule assembly within P-bodies. It is clear from their work that DDX6 granule size was decreased using the DDX6-RNase reporter. However, the specificity of the DDX6-RNase reporter is unclear, and clarification of this specificity would strengthen the authors findings. Overall, this work is interesting and provides important mechanistic insights into protein granule assembly and disassembly.

Authors response: We appreciate this positive assessment of our work!

Major comments:

- *The authors provide evidence for the mechanism of large DDX6 granule assembly throughout the paper. However, there are several figures where the characterization of the small and large granules is omitted. The division of granules based on size was ambiguous and there was no clear cut-off mentioned. To clarify the cut-off for small vs. large granules, please provide the total number, mean fluorescence intensity, and average number of small vs. large granules (Figure 1C-E and Figure 5C-F).*

Authors response: Thank you for pointing this out to us. We agree that the detailed analysis of granule parameters should be applied to all figures and have done so accordingly (see also point C of the common response part). Regarding the "lack of a clear cut-off": Our single particle analysis provides insight into the size and number of DDX6 granules across conditions, demonstrating **the variability in cell populations** and differences between conditions. Our initial analysis (e.g. **Fig. 1B, Fig. 2B**) was done by manually quantifying cell populations and visually classifying cells as either containing small or large DDX6 granules. The resulting data were presented as fraction of cells assigned to one of these two categories. To demonstrate how cells were assigned, we show examples of 22 DIV hippocampal neurons (**Fig. R1A**, see also revised **Supplementary Fig. 1A** of the main manuscript).

The analysis of single particle size (shown in **Fig. R1B**), results in two distinct pools of RNA granules based on age confirming our original findings. The dashed line in this panel represents the mean of the entire data set, which of course could serve as a defined cut-off requested by this referee. This cut-off could then be applied to calculate the percentage of large vs. small granules in each cell (**Fig. R1C**). As **such a cutoff substantially varied across all our experiments** due to variability between biological replicates, we would prefer not to include this in the main manuscript. This variability, however, **does not affect** the impact of our findings, as each experiment was assessed independently in comparison to control conditions within its biological replicates.

A

Figure R1: Classification of DDX6 granule size. **(A)** Representative examples of manual quantification of cell population. Cells classified as containing small granules are outlined in magenta, cells classified as containing large granules are outlined in green. **(B)** Density plot displaying two distinct pools of DDX6 granule sizes in 8 or 29 DIV hippocampal neurons in culture (related to **Fig. 1C-D**). Dashed line indicates mean of entire dataset. **(C)** Dot plot displaying average percent of large DDX6 granules per cell body in 8 or 29 DIV hippocampal neurons in culture, calculated using the mean of entire dataset as cut-off between small and large granules. **(D-E)** Dot plots displaying average mean fluorescent intensities of individual DDX6 granules in 22 DIV hippocampal neurons under vehicle treated or silenced (100 μ M CNQX, 50 μ M AP5, 1 μ M TTX) conditions (D, anti-DDX6 fluorescence) or in 14 DIV hippocampal neurons transfected with either GFP-DDX6 or GFP-DDX6-RNase1 reporters (E, GFP fluorescence). Distinct dot symbols indicate individual biological replicates. Small grey symbols represent single cells while larger white symbols indicate the average of each replicate. Horizontal line and error bars represent mean of replicates and standard deviation. Asterisks represent p-values obtained by Student's t-test (* $p < 0.05$, *** $p < 0.001$).

B

C

D

E

Finally, we would like to comment on the use of mean fluorescence intensity (MFI) as a parameter. As seen in **Supplementary Fig. 1E**, fluorescence signal intensity varied substantially between conditions. It was therefore necessary to adapt imaging parameters during acquisition at the microscope, to ensure adequate signal detection across all conditions. We would like to provide two representative

examples here in the rebuttal (**Fig. R1D-E, for the reviewers only**), where we kept imaging conditions comparable. In the first case, both average MFI (**Fig. R1D**) as well as particle size (**Fig. 2C**) increased; in the second, the average MFI remained largely unchanged (**Fig. R1E**) while particle size decreased (**Fig. 5C**). These two examples nicely illustrate that while granules can differ in size, the average protein concentration per area **must not necessarily change** as well. Of course, larger granules measured by DDX6 fluorescence will always contain more DDX6 protein in total. Consequently, we think that the size of granules is a more reliable marker than the mean fluorescence intensity.

- *Figure 5. The RNase reporter is active throughout the cell. Therefore, it is likely reducing total RNA content outside of P-bodies. The authors suggest that RNA availability is affecting the granule assembly within P-bodies. Could the authors show that the RNA within P-bodies* is also affected?*

Authors response: As discussed in point A of the common section (see above), we performed two new experiments to show that RNase1 activity outside granules does not account for the decrease in DDX6 granule size we observed in **Fig. 5**. Of course, we agree that it would be informative to show that RNA is affected directly in DDX6 granules.

First, we performed polyA FISH in the presence of our DDX6 or DDX6-RNase1 reporters (**Fig. R2A**). We quantified the fluorescence intensity of the polyA FISH signal overlapping with DDX6 granules (**Fig. R2B**). We observed a (modest) reduction of signal in the area of DDX6 granules. This experiment confirms RNase1 is active in the intracellular environment. Moreover, this data and the experiments of point A (see above and **Suppl. Fig. 5F-K**) suggest that the DDX6-RNase1 construct displays spatially restricted activity.

Second, we performed single molecule inexpensive fluorescence *in situ* hybridization (smiFISH) for three neuronally regulated mRNAs; *CamKII α* , *Rgs4* and *Calm3*. Unfortunately, all three targets showed only low levels of co-localization with DDX6 granules (**Fig. R2C**). Of course, various facts might account for this. First, we may have simply chosen the wrong mRNA candidate(s) for localization to DDX6 granules, as – at least for neurons – physiologically relevant and validated targets are **not yet known**. Second, RNA targets may be blocked from detection by FISH while packaged in granules, as shown by *Buxbaum et al., 2014 Science*, PMID: 24458642). Third, the three tested targets may in fact **only transiently** interact with DDX6 granules, a view that is supported by previous findings for protein interactors (Zeitelhofer *et al.*, 2008, *J. Neurosci.*, PMID: 18650333). In fact, the high turnover of granule components and the instability of RNA degradation products may make it difficult to specifically localize RNA depletion to granules vs. cytoplasm. With all those limitations mentioned, we nevertheless are convinced that detecting mRNA targets outside granules in the presence of our DDX6-RNase1 reporter is consistent with the notion that the tethered RNase1 **does not substantially affect** the entire pool of cytoplasmic mRNA (**Fig. R2D**).

Having shown that the RNase1 reporter is active in the cell and that its activity outside DDX6 granules does not interfere with DDX6 granule assembly (new **Suppl. Fig. 5F-K**), we now conclude that the activity of our GFP-DDX6-RNase1 reporter is responsible for the observed reduction in granule size, independent of its potential activity elsewhere. Identifying new suitable candidates for smiFISH in hippocampal neurons that localizes sufficiently to DDX6 granules would be substantial work that in our opinion would be clearly beyond the scope of these revisions. We hope that the reviewer will agree with us on that.

Figure R2: Fluorescence *in situ* hybridization in DDX6 or DDX6-RNase1 transfected hippocampal neurons. **(A)** Representative examples of polyA FISH in hippocampal neurons transfected with either BFP-DDX6 or BFP-DDX6-RNase1. **(B)** Dot plot displaying polyA FISH fluorescence intensity in areas overlapping BFP-DDX6 or BFP-DDX6-RNase1 signal. Distinct dot symbols indicate individual biological replicates. Small grey symbols represent single cells while larger white symbols indicate the average of each replicate. Horizontal line and error bars represent mean of replicates and standard deviation. **(C)** Dot plot displaying percent of DDX6 granules colocalizing with indicated smiFISH targets in neuronal cell bodies. Grey dots represent single cells while larger black dots indicate the average and error bars represent standard deviation. **(D)** Representative examples of *CamKII α* smiFISH in 22 DIV hippocampal neurons transfected with either BFP-DDX6 or BFP-DDX6-RNase1.

Minor comments:

- Figure 1B. The fraction of cells for 22 DIV has a negative error bar, which is not correct as a fraction cannot be negative. Perhaps the authors can either increase sample size or set SEM lower boundary to "0".

Authors response: Thank you for pointing this out. Indeed, we agree a fraction cannot be negative. The error bar displayed here, however, is **not** the SEM, but the standard deviation. Due to the high variability of the biological replicates, the error bars are quite large in this case, and are indeed negative. Of course, the actual fractions are not negative as can be seen by the individual data points we now display. We would prefer not to truncate the error bars as this would no longer represent the accurate standard deviation. However, if the reviewer wishes, we are happy to do so and would clearly state in the figure legend that error bars were shortened for the purpose of representation.

- *The authors always report large and small granules as a “fraction of cells.” Could the authors clarify what this means in the context of small vs. large granules? For example, do the authors mean the fraction of cells with large granules or the fraction of large granules per cell?*

Authors response: Thank you for the comment. We apologize for having been unclear. In these cases, we quantified the fraction of cells with large granules, by manually quantifying >100 cells at the microscope and categorizing them by either containing large or small granules as exemplified by the representative image we provide for each experiment. To clarify how these categories were assigned, we show examples of 22 DIV hippocampal neurons (**Fig. R1A**), which we now also include in the new **Supplementary Fig. 1A** of the main manuscript. We noticed early on in this study that granule size does not only vary within a cell, but that there is also considerable variation between cells. Therefore, we found it informative to include this information. We have now clarified this issue in the revised text.

- *Similarly, the fraction of cells has large variability. Could the authors please report the absolute number of granules (large and small) to show whether this variability is due to low numbers for a given cell?*

Authors response: We now included the absolute number of granules per cell for all experiments.

- *Figure 5C-F. Could the authors clarify whether the DDX6 granule quantifications based on the colocalization of all staining (GFP, DDX6 and DCP1a)?*

Authors response: The quantification is based on the GFP signal alone, as it functions as our reporter in this assay. We clarified this in the revised text.

- *Figure 1A, 2A, 3A and 4. The emphasis should be on the granules not on cell morphology. For figures with phase contrast images, either remove the phase contrast or place the inserts (zoomed in images) before the phase contrast images. Also, indicate/label examples of small and large granules in the zoomed insets.*

Authors response: We are not sure that we understand the concern. The phase contrast pictures are crucial to assess the physiological status of the cells as well as the RNA granules (vs. pathological condensates) and they were always placed below the respective images, which of course contain the important biological information. So, the images including the insets are always **before** the phase contrast pictures.

Furthermore, we now labeled small versus large granules in the inset of Fig. 1 A. We have adapted the Figures accordingly.

- *Figure 3. Include quantification of granule size and mean fluorescence. Also, include co-immunofluorescence of Stau2 and DDX6 to determine whether they are colocalized within the granules.*

Authors response: We now include the requested parameters after reanalyzing the data by single particle analysis. As imaging parameters had to be adapted to ensure adequate detection across conditions, the MFI is not a suitable parameter (as discussed above in the first major comment).

Moreover, we are happy to provide a representative example of DDX6 and Stau2 immunofluorescence. As can be seen by the images in **Fig. 3A**, there are only few instances where DDX6 and Stau2 colocalize in the cell body. This is in line with our previous publications showing that Stau2 transport granules and P-bodies are distinct entities, but show clear transient interactions (*Zeitelhofer et al.*, 2008, *J. Neurosci.*, PMID: 18650333) and that DDX6 is present in IP enriched Stau2 granules shown by mass spectrometry (*Fritzsche et al.*, 2013, *Cell Rep.*, PMID: 24360960).

Reviewer #2 (Remarks to the Author):

In this manuscript Bauer et al assess DDX6 granules formation during hippocampal neuron maturation, and investigate the role of RNA in this granule formation. First, the authors use hippocampal neurons to make the interesting observation that RNA granule size decreases as neurons mature in culture, and further observe similar correlations in vivo. The authors then demonstrate that inhibition of synaptic activity results in larger granules, whereas activation has a reciprocal impact. They use knockdown of Staufen 2 RNA binding protein to show that granule assembly following neuronal inhibition depends upon Staufen 2. Manipulating translation with inhibitors which release RNA also impacts granule size, as does mis-expressing DDX6-RNAase. These two experiments suggest that depleting RNA disrupts DDX6 granule formation.

The study presented by Bauer et al has some interesting findings, including the observation that RNA granules change during neuronal maturation and in response to synaptic activity. However, there are several concerns with the paper-as it feels somewhat preliminary, the flow of the paper somewhat disjointed and the motivation for specific experiments is often unclear. Furthermore, while the experiments are generally well rigorous there are some missing controls that affect the impact.

Authors response: We appreciate this productive and critical assessment of our work allowing us to add additional experiments, perform missing controls and explain the underlying motivation better. We feel that this really helped us improving the manuscript.

Main concerns:

1) Overall, the results in this paper are quite preliminary. While the observation that RNA promotes granule formation is interesting, this is not a new concept. Unfortunately, the impact of their findings alone are limited as they are very correlative. What is the functional impact of changing DDX6 granule size or showing that it depends upon RNA? Does granule size affect neuronal activation or synaptic function? Extending their study to other granules may also increase the impact of this study as would showing that it was functionally important for the cells. It sounds like one of the co-submitted papers may do this, and the authors may reconsider combining data as appropriate.

Authors response: As suggested, we performed trial experiments to investigate whether other types of RNA granules display similar assembly behavior as DDX6. To this end, we performed immunostainings for 6 different RNA-binding proteins (RBPs) involved in various cellular processes; e.g. ZBP1, UPF1, RBM14, Pur α , Pum2 and Mov10. We previously identified these RBPs (among other proteins, such as DDX6) as components of both Stau2 and Barentsz (Btz) granules in the rat brain by mass spectroscopy (*Fritzsche et al.*, 2013, *Cell Rep.*). All RBPs localized in defined granules in 8, 22, and 29 DIV hippocampal neurons during neuronal maturation (revised **Supplementary Fig. 1C**). All targets displayed a mostly fine granular pattern. None of the targets showed any clear changes in granule formation, comparable to DDX6, during neuronal maturation. Though we cannot exclude that the

assembly of these granules may be governed by similar mechanism(s) as DDX6, we feel a more detailed analysis is beyond the scope of this manuscript at the time. However, the data clearly shows that different RNA granule markers display distinct localization patterns. Moreover, DDX6 specifically undergoes substantial reorganization and changes in assembly during neuronal maturation compared to other RBPs (**Fig. 1A**).

The question whether the RNA dependent DDX6 granule assembly we present here has a functional impact on synaptic activity is of course of utmost interest. Past studies have clearly established that neuronal function is tightly linked to the regulation and expression of specific mRNAs (Mauger et al., 2016, *Neuron*, <https://doi.org/10.1016/j.neuron.2016.11.032>). Moreover, DDX6 and P-bodies have been extensively studied as translational repressors and regulators of mRNA (Corbet & Parker, 2019, *CSH Symp. Quant. Biol*, PMID: 32482896; Hubstenberger & Weil, 2017, *Mol. Cell*, PMID: 28965817). Therefore, it is plausible that DDX6 granules represent important regulatory entities in neurons, which might affect physiological processes, including neuronal function. This is supported by the fact that DDX6 is required for translationally regulated dendrite morphogenesis in the fly brain (Barbee et al., 2006, *Neuron*, PMID: 17178403). Interestingly, a manuscript from the lab of Florence Besse found that assembly of the DDX6 homolog (Me31B) affects the translatability of mRNAs during aging in the *Drosophila* brain (Pushpalatha et al., co-submitted/under revision).

To investigate whether DDX6 may be involved in the regulation of synaptic function in our system, we knocked down DDX6 in hippocampal neurons, by lentiviral transduction of a short hairpin RNA targeting *DDX6* mRNA (shDDX6). The shDDX6 used here resulted only in a modest reduction of DDX6 protein and a reduction of DDX6 granule size, mimicking the reduction in protein levels and granule size observed during neuronal maturation (**Fig 1A-C and Supplementary Fig. 1E-F**). This knock down resulted in an increased clustering of the postsynaptic marker Homer1 in dendrites of 15 DIV hippocampal neurons (**Fig. R3A-B**). Moreover, Ca²⁺ live imaging displayed an increase in the frequency of fluorescent peaks in neurons where DDX6 was depleted (**Fig. R3C-D**). Together, this data indicates, that DDX6 is involved in the regulation of synaptic function. Considering the role of DDX6 as a translational repressor and the regulation of its assembly we present in our manuscript, we propose that DDX6 granule assembly may repress specific synaptic target mRNAs. In line with this, depletion of DDX6 may de-repress such targets, resulting in de-regulated synaptic formation and activity. However, clearly more work beyond the scope of this paper is necessary to prove that this is indeed the case.

Figure R3: DDX6 depletion alters synaptic function. **(A)** Representative examples of anti-Homer1 immunofluorescence in dendrites of 14 DIV hippocampal neurons in culture transfected either with shNTC or shDDX6. Scale bar 10 μm . **(B)** Bar plot displaying average Homer1 cluster size in 20 μm segments binned along dendrites. Error bars = SEM. **(C)** Representative examples of Ca^{2+} sensor fluorescence intensity over time in 15 DIV hippocampal neurons transduced with shNTC or shDDX6. **(D)** Dot plot displaying average number of fluorescence intensity peaks per minute. Distinct dot symbols indicate individual biological replicates. Small grey symbols represent single cells while larger white symbols indicate the average of each replicate. Horizontal line and error bars represent mean of replicates and standard deviation.

2) The **quantification of experiments and description** could be improved. First, there were insufficient details provided for how the granules were classified as large or small. Please provide exact cut-offs and rationale for these cut-offs. (For example in Figure 1C there are about 12% of granules which are large (fig 1B) but how does that equate to data in Fig 1C? Is it the few outliers? How was this selected?) Second, the data in Figure 1F reflect a shift to more cytoplasmic signal of DDX6. To what extent does this reflect **more cell area** (differences in size of older neurons?)

Authors response: Thank you for these interesting comments. We apologize for not having been clear enough on how our analysis was performed. Please also see our response to Referee #1 (page 4-5). We have now revised the text and provide more experimental detail.

In short, cells were manually classified as either containing large or small granules as shown by the representative images we provide (see Fig. R1). To clarify how these categories were assigned, we show examples of 22 DIV hippocampal neurons (Fig. R1A), which is also included in Supplementary Fig. 1A of the main manuscript. As this is a qualitative analysis done manually, a defined cutoff is not possible (but see Fig. R1B). However, as >100 cells were quantified, this data gives meaningful insight into the variability in cell population. Additionally, our new single particle analysis provides more

detailed insight into granule parameters. Therefore, while **Fig. 1B** provides a general assessment of the cell population, **Fig. 1C** gives the actual average granule size per cell in μm^2 .

Concerning **Fig. 1E**, it is true that older cells indeed are larger than younger cells (**Fig. R4A**). However, the ratio in **Fig. 1E** was calculated using the mean fluorescence intensity (total fluorescence intensity/number of pixels), which is independent of the area quantified. Alternatively, the total integrated fluorescence intensity may be used (**Fig. R4B**), which shows the same effect. However, this does indeed **not** account for cell size.

Figure R4: Dot plot displaying cell body size (**A**) and granular to cytoplasmic integrated fluorescence intensity ratio (**B**) in hippocampal neurons at 8 and 22 or 29 DIV, respectively. Distinct dot symbols indicate individual biological replicates. Small grey symbols represent single cells while larger black symbols indicate the average of each replicate. Horizontal line and error bars represent mean of replicates and standard deviation. Asterisks represent p-values obtained by Student's t-test ($**$ $p \leq 0.01$, $***$ $p \leq 0.001$).

3) A major point of the paper is that RNA availability affects granule size, but this observation is limited to analyses at one stage (either with translational inhibitors or with GFP-DDX6-RNase1). Are the changes in granule size seen during neuronal maturation (Fig 1) and neuronal activity (Fig 2) also driven by RNA? The role of RNA in granule formation in these contexts could be further fleshed out to increase the impact of their findings.

Authors response: Thank you for this interesting comment. We agree that this is a very crucial question and addressing it would substantially solidify our findings. Due to the impact of this important point, we have presented it in point B of the common section above. In brief, we found that DDX6 granule assembly upon neuronal inhibition requires RNA supply in mature hippocampal neurons (**Fig. 5E and Supplementary Fig. 5L-M**). We think that this finding clearly links RNA dependent DDX6 assembly to neuronal function. Furthermore, this is in line with our initial findings observed during neuronal maturation (**Fig. 1**), as synapse formation occurs and results in increased synaptic activity. The establishment of the synaptic network may therefore be responsible for the observed reorganization of DDX6 granules during neuronal maturation. In contrast, the inhibition of synaptic activity results in RNA dependent granule assembly, reversing the effects of maturation on DDX6 granule formation in general terms.

To inquire whether changes in granule size during neuronal maturation are also dependent on RNA, we analyzed the effect of our RNase reporter system in immature (8 DIV) neurons in culture (**Fig. R5**) in addition to the data presented in **Fig. 5**. We found a reduction in GFP-DDX6-RNase1 granule size compared to the GFP-DDX6 reporter (**Fig. R5A**). Granule number and mean fluorescence intensity did not show a reproducible effect, between the two replicates preformed (**Fig. R5B-C**). However, the reduction in granule size indicates that DDX6 assembly in immature neurons is dependent on RNA, to a similar extent as observed in mature neurons. This would suggest, DDX6 granule formation is governed by RNA supply during multiple stages of neuronal maturation, including immature neurons that have not yet developed a functional synaptic network. Therefore, while RNA supply can be seen

as a general mechanism for granule assembly, our data indicates that the onset of synaptic activity in mature neurons would represent an additional upstream mechanism of regulation.

Figure R5: RNA degradation in DDX6 granules of immature 8 DIV hippocampal neurons results in disassembly. (A-C) Dot plots displaying average DDX6 granule size (A), granule number per cell body (B) and average fluorescence intensity (C) in hippocampal neurons transfected either with GFP-DDX6 or GFP-DDX6-RNase1 reporters. Distinct dot symbols indicate individual biological replicates. Small grey symbols represent single cells while larger white symbols indicate the average of each replicate. Horizontal line and error bars represent mean of replicates and standard deviation.

Likewise, the observation that silencing synaptic activity leads to larger granules is interesting, especially since this is more similar to an immature neuron. This begs the question of how granule formation changes with maturation - are the granules less mobile when they are larger or in response to silencing synaptic activity? Fleshing their observations out on this front, with live imaging to assess the mobility of granules of different sizes, or understand if granules coalesce with maturation or grow larger from independent granules, could also be interesting.

Authors response: To inquire whether granules coalesce or grow larger from independent granules during DDX6 assembly, we performed live imaging of GFP-DDX6 (**Supplementary Fig. 1H and movie 1**). We observed individual DDX6 granules fusing and splitting over time, which is in line with the behavior of similar granules reported in cell (Folkmann et al., 2021, *Science*, PMID: 34516789; Sankaranarayanan et al., 2021, *Dev. Cell*, PMID: 34655524). Next, we induced granule disassembly by the application of NMDA and subsequently permitted assembly by NMDA wash-off (**Supplementary Fig. 2B and movie 2**). Granules quickly disassembled upon NMDA treatment and reassembled upon wash-off. We observed individual granules growing smaller or larger directly, without splitting or fusion. It is plausible that the fast disassembly and reassembly observed here occurs by modulating existing granules, rather than splitting/fusion or the formation of new condensates.

The nature of those *live-cell* imaging experiments does not enable us to draw conclusions on the behavior of granules **during longer processes such as neuronal maturation** as it does not allow us to follow single granules during maturation to identify the predominant mechanism of regulation. Therefore, the current data set does not provide such detailed molecular insight into either assembly frequencies or mobilities as suggested by this referee. Though clearly an interesting question, it is beyond the scope of this revision. However, we are happy to include our preliminary findings if this would be requested.

4) Given the difference in granule size in response to changes in synaptic activity, it would be important to look at granules **sub-cellularly, at the synapse or in dendrites specifically** (Fig 2). Likewise, as *Staufen2* is important in mRNA localization and particularly to DDX6 granules, it would be valuable to assess how *Stau2* KD affects granules at localized compartments (i.e.: soma versus synapse and/or dendrites).

Figure R6: Stau2 depletion inhibits DDX6 granule assembly upon neuronal inhibition in dendrites. **(A)** Representative examples of DDX6 immunostainings in dendrites of shNTC and shStau2 transduced 22 DIV hippocampal neurons in culture either under vehicle (DMSO) treated or silenced (100µM CNQX, 50µM AP5, 1µM TTX) conditions. Scale bar 10 µm. **(B-C)** Bar plots displaying average DDX6 granules size (A) and average DDX6 granule number (B) in 10 µm bins along the first 50 µm of dendrites. Data represents mean ± standard error of the mean of one experiment.

Authors response: Thank you for this interesting question. Though the focus of our study is the regulation of DDX6 granules located in the cell body, we agree that exploring the distinct regulation in other compartments would be relevant. This is of course particularly interesting for dendrites, which may give insight into neuronal function. Therefore, we investigated DDX6 granules in dendrites upon neuronal inhibition and in the presence or absence of Stau2 protein (**Fig. R6**). We found DDX6 granules prominently localized in dendrites in all conditions (**Fig. R6A**). Next, we quantified DDX6 granule size and number, and binned the data in 10 µm segments along the first 50 µm of dendrites to gain detailed insight into their distribution (**Fig. R6B-C**). Though granules were generally smaller in distal dendrites (e.g. segment 0-10 vs. 40-50), neuronal inhibition caused a small increase in average DDX6 granule size in all segments under control (shNTC) conditions (**Fig. R6B**). Interestingly, Stau2 depletion (shStau2) prevented the assembly of larger granules upon neuronal inhibition, clearly reproducing our results obtained from the cell body (**Fig. 3B-D**). We observed no clear change in DDX6 granule number in any conditions (**Fig. R6C**). This is in line with our previous data, showing that neuronal inhibition does not affect the number of DCP1a granules (another P-body marker) in dendrites (Zeitelhofer *et al.*, 2008, *JNS*, PMID: 18650333). In conclusion, this data supports our original finding. Moreover, it indicates that DDX6 granule assembly is regulated locally in dendrites in addition to the cell body. This may enable translational regulation of localized transcripts in response to synaptic cues.

5) *The motivation for manipulating Staufen2 was unclear. Perhaps the co-submitted paper would help clarify, but in the context of this manuscript, it remains unclear.*

Authors response: Thank you for pointing this out. We have previously identified DDX6 (RCK) as an interaction partner of Stau2 in neurons (*Fritsche et al., 2013, Cell Rep.*, PMID: 24360960). Furthermore, we have previously shown that both proteins interact transiently and that DDX6 occurs both in P-bodies as well as in neuronal transport granules (*Zeitelhofer et al., 2008, JNS*, PMID: 18650333). Moreover, the double-stranded RBP Stau2 is an important RNA transporter in neurons, which recognizes complex folded RNA secondary structures in the 3'-UTR of its target RNAs (*Tang et al, 2001, Neuron*, PMID: 11709157; *Fernandez-Moya et al, 2021, Intl. J. Mol. Sci.*, PMID: 34884825). This points towards a regulatory role of RNA structures in the context of Stau2 knock-down and that Stau2 is likely to regulate RNA supply inside neurons. Consequently, a DEAD box RNA helicase such as DDX6 might play a role in the regulation of such complex targets. Therefore, we investigated whether manipulating Stau2 would impact DDX6 granules in neurons. We clarified our rationale in the revised text.

6) *The translation inhibitor experiments could be better developed. First, it is unclear why they only use CHX under silenced conditions. The comparison with PMY and HRN at baseline would be important for interpreting their findings.*

Authors response: Thank you for bringing this up. We agree with this referee that comparing PMY, HRN with CHX treated cells under basal conditions is important. We therefore treated mature hippocampal neurons (22DIV) with CHX for 4 h. Interestingly, we observed only a moderate decrease in DDX6 granules size upon ribosome stalling with CHX, likely due to the fact that granules are already quite small in size at 22 DIV (revised **Supplementary Fig. D-E**). We therefore, repeated the treatment in 8 DIV neurons, where granules are larger at baseline, and found a clear reduction in size (**Fig. R7A**). This finding is underlined by our observation that polysome destabilizing drugs such as PMY and HRN increase DDX6 granule size at baseline (**Fig. 4A-F**). Together, our results are in line with our hypothesis that the levels of free, cytosolic RNA availability crucially regulates DDX6 granule assembly. To better assess the effects of CHX at 22 DIV and to inquire whether it acts upstream or downstream of neuronal activity, we first inhibited neuronal activity to induce large granules and subsequently applied CHX as shown in **Fig. 4G-H**. We adapted the text accordingly.

Figure R7: DDX6 granule disassembly is facilitated by incubation with cycloheximide (CHX). **(A)** Bar plot displaying quantification of cell population by fraction of cells containing either large or small DDX6 granules in 8 DIV hippocampal neurons. **(B)** Immunoblots of cortical neurons treated with PMY (25 μ M 15 min), CHX (7 μ M 4 h), HRN (2 μ g/mL 30 min) or vehicle, decorated against PARP, b-III Tubulin and Casp3. **(C)** Quantification of relative fluorescence intensity of PARP and Casp3 normalized to loading control from Western blots of three biological replicates.

Abbreviations:

PMY=puromycin, HRN=harringtonine, CHX=cycloheximide.

Second, translation inhibitors will do a lot more to the cell than simply release RNA. How can the authors rule out that observed differences are not due to some other impact of inhibitors?

Authors response: We agree with this referee that translation inhibitors can have a broad impact on the proteome due to a decrease in general translation activity. Several lines of evidence, however, speak against an indirect effect of the inhibitors on DDX6 granules. First, even though PMY, HRN and CHX inhibit translation, we observed different effects on DDX6 granule size suggesting that general shut down of translation activity is not the sole driving force. Second, according to published mass spectrometry results, neuronal proteins have high stability resulting in a half-life time of days (Dörrbaum *et al.* 2020 *eLife*, PMID: 32238265). Therefore, a broad impact on the proteome seems to be unlikely. And third, our polysome profiling experiments show that DDX6 granules do not interact with initiating or translating ribosomes. In contrast to other RBPs such as FMRP (Darnell *et al.* 2011, *Cell*, PMID: 21784246; Stefani *et al.* 2004, *J. Neurosci.*, PMID: 15317853) that associate with translating ribosomes to sense translational activity (Shu *et al.* 2020 *PNAS*, PMID: 33199649; Chen *et al.* 2014 *Mol. Cell*, PMID: 24746697), we think it is more likely that DDX6 granule assembly is regulated by RNA availability rather than indirectly by the ribosome.

To further address indirect effects that might be caused by translation inhibitors, we analyzed expression levels of two different apoptotic markers, PARP and Caspase-3. Importantly, we did not observe any significant differences in steady-state levels of PARP and Caspase-3 expression upon incubation of mature cortical neurons with PMY, HRN or CHX (**Fig. R7B-C**). This finding indicates that apoptosis is not induced upon translation shut down ruling out that cellular stress is the driving force that controls DDX6 assembly.

7) The tethering of RNase1 to GFP-DDX6 affects total RNA availability (Fig. 5 and S5), which then influences DDX6 granule size. This assay needs some additional controls. How can the authors be confident that this tag does not affect DDX6 localization in the cell? - the pattern of GFP looks different and this needs to be assessed. It would be valuable to also include an experiment where RNase1 is tethered to a non-RBP or an RBP found in a distinct granule (ie: stress granules; perhaps G3BP1) to show specificity of this finding.

Authors response: Thank you for providing these important points. We fully agree. To test whether the RNase1 tag affects localization and is responsible for the observed change in DDX6 granule size, we mutated its catalytic site. Specifically, two histidines (H40 and H147, referring to the full-length rat RNase1 precursor) were targeted and replaced by the aliphatic amino acid leucine to inactivate RNase1 (Fig. R8A). We termed this mutant RNase1-HL. In an *in vitro* activity assay, the GFP-DDX6-RNase1 reporter showed clear RNA degradation, while the mutated reporter GFP-DDX6-RNase1-HL showed little to no degradation activity (Fig. R8B). Moreover, GFP-DDX6-RNase1-HL localized comparably to GFP-DDX6 and GFP-DDX6-RNase1 when expressed in hippocampal neurons (Fig. R8C). Interestingly, the mutation of the RNase1 tag partially rescued the effects of wild type RNase1 as GFP-DDX6-RNase1-HL granules were larger than GFP-DDX6-RNase1 granules. This experiment indicates that the tag alone does not account for the observed phenotype, but requires RNase1 activity. Though in our opinion this is a very interesting result, a more detailed analysis is needed.

We agree that experiments where RNase1 is tethered to locations distinct to DDX6 granules would be very valuable in assessing the specificity of our findings. As we consider this an essential point, we chose to discuss this in point A of the common section above. In summary, we found that both untethered RNase1, freely diffusing in the cytoplasm, or tethered to mitochondria did not affect GFP-DDX6 granules size, number or fluorescence intensity. Therefore, RNase activity outside DDX6 granules does not account for the observed phenotypes in Fig. 5.

Figure R8: Mutating the catalytic site of the RNase1 tag results in decreased degradation activity and rescues DDX6 granule assembly. (A) Scheme of GFP-DDX6, GFP-DDX6-RNase1, and GFP-DDX6-RNase1-HL expression cassettes. (B) Formaldehyde agarose gel stained with ethidium bromide upon incubation of isolated total rat brain RNA with either GFP-DDX6, GFP-DDX6-RNase1 or GFP-DDX6-RNase1-HL. (C) Representative examples of GFP fluorescence in 14 DIV hippocampal neurons in culture, transfected with either GFP-DDX6, GFP-DDX6-RNase1 or GFP-DDX6-RNase1-HL reporters. Boxed regions in images are displayed as magnified insets.

Minor concerns:

1) *The authors may consider using super-plots to represent their data (Lord, SJ, et al, JCB 2020).*

Authors response: Thank you for this suggestion. We find this representation very attractive and informative. Therefore, we adapted our plots where possible and included single data points for each biological replicate in all graphs.

2) *It is unclear whether the N for the statistics is # of cells or independent experiments. Using the number of cells quantified for statistical tests is inappropriate.*

Authors response: We apologize for having been unclear. For our single particle analysis, statistics were indeed performed on single cells as indicated by the plots. Our impression is that this is the way statistics are handled in most publications in the field at the current time. Treating individual cells as replicates of course makes statistics more powerful as more data points are included. However, we agree that this does not consider variations in each cell culture and may hide whether data is reproducible between cultures. Therefore, we have adapted our statistical analysis for these cases, and clarified this in the Methods section.

3) *In Fig 2C, the baseline fraction of cells with large granules is different than in Fig 1 B (40% v 25%)*

Authors response: This is indeed true. In fact, the high variability of our cell culture system (*e.g.* differences in the rate of maturation or embryo development) was a particular challenge for this study. The fraction of cells containing larger or smaller granules varied substantially between biological replicates, which is why we took special care to include control conditions for all experiments. This is particularly obvious in **Fig. 1B**, as pointed out by the reviewer. We hope that the display of individual data points, which we included in the revised plots makes this issue clearer. However, this variability is simply due to our model systems and is particularly obvious for DDX6 granules, only supporting the notion that these condensates strongly depend on neuronal maturation and function.

4) *Need references for statement that synaptic activity determines localization and translatability of mRNAs (figure 3)*

Authors response: Thank you for pointing this out. We included suitable references in the revised manuscript.

5) *They show that Stau2 is required for formation of large granules after silencing activity. Is the reverse true? Does Stau2 overcome formation of small granules following activation?*

Authors response: This is an interesting question. To address this, we transiently overexpressed either RFP or RFP-Stau2 in mature hippocampal neurons overnight and subsequently stimulated neuronal activity via the application of 100 μ M NMDA for 15 minutes (**Fig. R9A**). NMDA treatment was able to effectively disassemble DDX6 granules both in the presence or absence of Stau2 overexpression. We hypothesized that the depletion of Stau2 may in turn affect granule disassembly upon neuronal activity. Therefore, we induced neuronal activity by NMDA in neurons where Stau2 was knocked-down by an shRNA (**Fig. R9B**). However, NMDA was able to induce DDX6 granule disassembly similarly in the

presence or absence of Stau2. These preliminary experiments indicate that Stau2 does not have a role in NMDA mediated DDX6 disassembly. Considering that NMDA induced disassembly occurs in a shorter timeframe than assembly by neuronal inhibition, we suspect the underlying mechanisms may be different. It is in line with neuronal function that stimulation would cause the release of relevant mRNA targets from DDX6 granules for transport and possibly local translation in dendrites near synapses. As we have recently shown, Stau2 plays an essential role in transporting its target mRNA to synapses upon synaptic stimulation (Bauer *et al.*, 2019, *Nature Comm.*, PMID: 31320644). Therefore, we propose that Stau2 does not directly pull mRNA out of DDX6 granules, but may bind released mRNA after DDX6 granule disassembly. In turn, the assembly of DDX6 granules upon neuronal inhibition, which occurs in a much longer timeframe over hours, would require the supply of mRNA by Stau2.

Figure R9: Neuronal stimulation regulates DDX6 disassembly upstream of Stau2. **(A)** Representative examples of DDX6 immunofluorescence and RFP fluorescence in 16 DIV hippocampal neurons transfected with either RFP or RFP-Stau2 upon 15 min vehicle or 100µM NMDA treatment, respectively. **(B)** Representative examples of DDX6 immunofluorescence in shNTC or shStau2 transduced 18 DIV hippocampal neurons upon 15 min 100µM NMDA or mock treatment, respectively.

6) Use of word in vivo needs to be tempered in the conclusions, as all experiments besides immunofluorescence in Fig 1 are in vitro experiments

Authors response: Thank you for pointing this out. We adapted the manuscript accordingly.

Reviewer #3 (Remarks to the Author):

The manuscript by Bauer et al. examines how the availability of RNA shapes DDX6 granule condensation using an in vivo system of maturing neurons in culture. The authors begin by describing a gradual de-condensation of DDX6 granules upon neuronal maturation, both in cultured neurons and mice brain sections. They then show that DDX6 granules can assemble in response to synaptic inhibition in a reversible manner. This inhibition-induced assembly is sensitive to Stau2 knockdown (with the knockdown alone not affecting DDX6 granule size). The authors then compare three translation inhibitors that differentially affect the pool of ribosome-associated RNAs and provide evidence that the availability of ribosome-free cytoplasmic RNA is a determinant of granule condensation. Finally, the authors use a DDX6-RNase1 fusion to locally deplete RNA from granules and observe a reduction in granule size further demonstrating the importance of RNA in granule condensation.

Investigating the relationship between DDX6 granule dynamics and RNA availability in a physiological context is interesting and adds new insight to the field of granule biology. This manuscript is well written, concise, and easy to follow. Experiments are nicely controlled and clearly presented. The paper's conclusions are overall properly supported by the data.

Authors response: We appreciate this very positive assessment of our work!

I have one major comment that needs to be addressed before recommending the manuscript for publication: To directly demonstrate the effects of RNA depletion in granules, the authors induce localized RNA degradation through a GFP-DDX6-RNase1 fusion. My concern here is that a fraction of GFP-DDX6-RNase1 does not localize in granules (as demonstrated by the diffuse cytoplasmic GFP and IF signals in Fig. 5B for instance, and the mean fluorescence granular/cytoplasmic ratio in Fig. 1E). This cytosolic non-granule fraction of GFP-DDX6-RNase1 will therefore deplete non-granule RNA. It will thus improve this manuscript to prove that this system preferentially degrades RNA in granules. To demonstrate this, I suggest (i) adding smFISH experiments quantifying a granule-enriched and a non-enriched transcript with GFP-DDX6 or GFP-DDX6-RNase1 expressed, (ii) and/or comparing the measured granule parameters with an additional construct containing RNase but not fused to DDX6 or any granule marker. These experiments will consolidate the notion that RNAs, specifically in granules, are a determinant of condensation.

Authors response: Thank you for these great suggestions. We fully agree. We consider the question whether RNase1 activity degrades RNA preferentially in granules essential. This is why we already discussed this in point A of the common section above. In summary, we found that both untethered RNase1, freely diffusing in the cytoplasm, or tethered to mitochondria did not affect GFP-DDX6 granules size, number or fluorescence intensity (**Supplementary Fig. 5F-K**). Therefore, RNase1 activity outside DDX6 granules appears not to account for the observed phenotypes in **Fig. 5**. Furthermore, we agree that demonstrating RNA depletion in GFP-DDX6-RNase1 granules directly would be valuable. As reviewer 1 made a similar inquiry, we already discussed experiments performed during the revisions in the second comment of reviewer 1 and Fig. R2 above.

Regarding the suggestion to test for RNA degradation in granules: First of all, as discussed above (**Fig. R2A-B**), we were able to show a reduction of RNA close to DDX6 granules by polyA FISH. Unfortunately, there are no validated DDX6 mRNA targets known in neurons to assay for specific targets. Potential reasons may be that our selected targets do in fact not localize to DDX6 granules or that they are inaccessible for FISH while packed in granules, as shown by Buxbaum *et al.*, 2014, *Science*, PMID: 24458642). In any case, the identification of suitable neuronal DDX6 targets would require substantial work we feel is beyond the scope of the current revisions. We hope that this reviewer will agree with us on this. To summarize, we show that RNase1 is active in the intracellular environment (**Fig. R2A-B**) and that RNase1 activity outside granules does not account for the reduction in DDX6 granule size observed in **Fig. 5** (**Supplementary Fig. 5F-K**). Therefore, we believe that it is fair to conclude that the activity of our GFP-DDX6-RNase1 reporter is responsible for the observed reduction in granule size, independent of its potential activity elsewhere.

Minor comment: To provide a more complete view on the role of RNA availability in granule formation, please mention studies performed using artificial granules such as PMID: 31324804.

Authors response: Thank you for pointing this out. Of course, we are now mentioning such studies including the paper by Navarro *et al.*, 2019, *Nature Comm.*!

REVIEWERS' COMMENTS

Reviewer #1 (Remarks to the Author):

In the revised manuscript by Bauer et al, the authors have included additional information and have improved the analyses and overall presentation of their data. Although the major concerns were addressed, there is one lingering question that perhaps needs additional clarification.

While I believe the data indicate that DDX6 can bind RNA and partake in granules, the results do not unequivocally show that DDX6 is a regulatory component of granule assembly and disassembly. We know that RNA is not 'naked' inside the cell and are bound to numerous sequence non-specific RNA-binding proteins that protect the RNA from degradation. I believe the authors need to address the alternative interpretation of their results in the discussion that DDX6 may be passively binding non-specifically to "free" RNA. This function, in and of itself, can be potentially interesting but from the data it is difficult and premature to conclude that DDX6 acts as a kind of rheostat for granule dynamics in neurons.

Reviewer #2 (Remarks to the Author):

In this revised manuscript by Kiebler and colleagues, the authors have substantially improved the manuscript. They have been very attentive to original concerns which they have addressed with inclusion of several new experiments, new data analysis and improved data presentation, and modifications to the text-including improved rationale. They include new investigations comparing DDX6 localization to other RNA binding proteins. They also include new live imaging of granules. I also very much appreciate the important controls with new RNase reporters that do not localize to DDX6 granules. These show the RNase alone does not behave like the DDX6 targeted version, and show that assembly is controlled by synaptic activity. This valuable point is added to the abstract.

While I commend the authors for their detailed responses and attention to concerns raised, I did find it surprising that some of their new data were not integrated into the manuscript but rather included only in the response to reviewers. Mainly, of my main concerns was the lack of any functional investigation for how DDX6 granules affect hippocampal neurons. In Figure R3, the authors knocked down DDX6 and showed that this led to alterations in synaptic function (monitoring calcium imaging and postsynaptic marker localization). I feel these data may be valuable to include in the supplemental material as they lend some further functional significance to DDX6 and granules.

Overall, I am satisfied with the authors' responses to concerns and feel it makes valuable contributions to our understanding of RNA granules in neuronal maturation.

Reviewer #3 (Remarks to the Author):

The authors did a commendable job in revising their manuscript. The DDX6-RNase assay in particular is now more convincing and better controlled. The authors also shed light on and further developed several other aspects that make the manuscript richer. I recommend this paper for publication.

Minor comments:

- Comparing Figures 5C, D and S5I, J; the mean values of DDX6 granule size and number per cell body are rather different: smaller, more numerous granules in the new set of experiments using control RFP and TOMM20 +/- RNase constructs. The conclusion regarding the spatial specificity of RNase activity still holds. However, I am curious as to where this difference comes from. Is less DDX6-GFP co-transfected in these conditions?
- It would be clearer if the co-transfected DDX6-GFP construct is also indicated on the X-axis of the graphs in Figure S5I-K.

We would like to thank the reviewers for their positive assessment of our revised manuscript and the revisions in general. We are pleased that the reviewers appreciated our efforts to improve our manuscript according to their suggestions, and that the reviewers generally agree with our assessment of the data. Of course, we are happy to address the remaining comments of the reviewers below, where you will find the original reviewer comments in black and our author responses in blue text.

Reviewer #1:

In the revised manuscript by Bauer et al, the authors have included additional information and have improved the analyses and overall presentation of their data. Although the major concerns were addressed, there is one lingering question that perhaps needs additional clarification.

While I believe the data indicate that DDX6 can bind RNA and partake in granules, the results do not unequivocally show that DDX6 is a regulatory component of granule assembly and disassembly. We know that RNA is not 'naked' inside the cell and are bound to numerous sequence non-specific RNA-binding proteins that protect the RNA from degradation. I believe the authors need to address the alternative interpretation of their results in the discussion that DDX6 may be passively binding non-specifically to "free" RNA. This function, in and of itself, can be potentially interesting but from the data it is difficult and premature to conclude that DDX6 acts as a kind of rheostat for granule dynamics in neurons.

Thank you for pointing this out. It was not our intention to suggest DDX6 is a direct regulator of RNA granule assembly. Indeed, our data do not address this, as the use of DDX6 was intended to observe granule assembly processes in primary neurons. Recent literature by the Weis lab among others, however, provide evidence that DDX6 (Dhh1 in yeast) regulates RNP assembly and turnover (Hondele & Weis, 2019, Nature, PMID: 31435012). We carefully reassessed our manuscript and changed wording accordingly. Moreover, we mention the possibility that DDX6 acts as a non-specific free RNA binder in the discussion.

Reviewer #2:

In this revised manuscript by Kiebler and colleagues, the authors have substantially improved the manuscript. They have been very attentive to original concerns which they have addressed with inclusion of several new experiments, new data analysis and improved data presentation, and modifications to the text-including improved rationale. They include new investigations comparing DDX6 localization to other RNA binding proteins. They also include new live imaging of granules. I also very much appreciate the important controls with new RNase reporters that do not localize to DDX6 granules. These show the RNase alone does not behave like the DDX6 targeted version, and show that assembly is controlled by synaptic activity. This valuable point is added to the abstract.

While I commend the authors for their detailed responses and attention to concerns raised, I did find it surprising that some of their new data were not integrated into the manuscript but rather included

only in the response to reviewers. Mainly, of my main concerns was the lack of any functional investigation for how DDX6 granules affect hippocampal neurons. In Figure R3, the authors knocked down DDX6 and showed that this led to alterations in synaptic function (monitoring calcium imaging and postsynaptic marker localization). I feel these data may be valuable to include in the supplemental material as they lend some further functional significance to DDX6 and granules.

Overall, I am satisfied with the authors' responses to concerns and feel it makes valuable contributions to our understanding of RNA granules in neuronal maturation.

Thank you for your kind suggestions. We were excited about our new data addressing the impact on neuronal function as well. However, as the data is yet somewhat preliminary in our opinion, we decided not to include it in the main manuscript. However, following the suggestion of this reviewer we now include this data in the supplementary materials and added a suitable new passage to the manuscript. Furthermore, we now include a movie providing a representative example of calcium imaging in living neurons, for the readers.

Reviewer #3:

The authors did a commendable job in revising their manuscript. The DDX6-RNase assay in particular is now more convincing and better controlled. The authors also shed light on and further developed several other aspects that make the manuscript richer. I recommend this paper for publication.

Minor comments:

- *Comparing Figures 5C, D and S5I, J; the mean values of DDX6 granule size and number per cell body are rather different: smaller, more numerous granules in the new set of experiments using control RFP and TOMM20 +/- RNase constructs. The conclusion regarding the spatial specificity of RNase activity still holds. However, I am curious as to where this difference comes from. Is less DDX6-GFP co-transfected in these conditions?*

Thank you for pointing this out. Indeed, less DDX6-GFP was transfected in experiments where the reporter was co-transfected together with the new RFP reporters. In this experiment, two plasmids were co-transfected in each condition, however the total amount (3 µg) was kept the same as in single plasmid transfections, as per our protocol. We added a clarification in the methods section. Additionally, biological variability may be a relevant factor in this case. Differences in expression levels of the transiently expressed plasmid due to technical or biological variability may indeed lead to variances in the detection of granules (*e.g.* due to different fluorescent background levels). Therefore, we made sure to include proper controls for each replicate, and compare the data within each primary neuronal culture.

- *It would be clearer if the co-transfected DDX6-GFP construct is also indicated on the X-axis of the graphs in Figure S5I-K.*

Thank you for this suggestion. We agree and added this to the top of the plot, to be consistent with other figures throughout our manuscript.